# Loss avoidance during social interactions

Benjamin J. Kuper-Smith [1,2] ✉ & Christoph W. Korn [1,2] ✉

Social interactions lead to outcomes for oneself and others, which can be gains or losses. Yet, it is unclear how exactly people's social decisions are affected by whether an outcome is above or below zero. We systematically varied whether the outcomes of social dilemmas (Prisoner's Dilemma, Stag Hunt, Chicken) were gains, losses, or combinations thereof. Across seven experiments (4 preregistered; $N_{Offline} = 197$, $N_{Online} = 1653$), participants tried to avoid losses altogether (loss avoidance), but there was no consistent evidence that they tried to minimize losses (loss aversion). If cooperation avoided losses, people cooperated more; if defection avoided losses, people defected more, even if this imposed a loss on the other person. Our results suggest that cooperation and social interactions can be influenced systematically if the situation allows people to avoid losses.

Human life contains countless decisions that affect oneself and other people. Whether a salary negotiation, splitting a bill in a restaurant, or investors contributing to a joint business venture, the outcomes of such decisions can be gains, losses, or some combination thereof for some or all people involved. Does it make a difference whether the outcomes of such decisions are positive or negative? People prefer gains to losses, but losses and gains are not necessarily opposite sides of the same continuum: a vast literature across different situations suggests asymmetric decision-making between positive and negative outcomes[1–5], including different neural mechanisms[6–8]. One of the most influential theories of economic choice, prospect theory[9], includes an asymmetry between gains and losses: loss aversion[9–11] suggests that people weigh losses more than they weigh equivalent gains.

Loss aversion has received most of the attention in the discussion of asymmetries between losses and gains, but another asymmetry, loss avoidance, has received much less attention. Loss aversion is typically defined as a steeper slope for losses than for gains, while loss avoidance[12,13] is typically defined by an intercept for losses, applying a 'utility fine' to any loss, irrespective of magnitude. Loss aversion and loss avoidance have often been conflated, so it is unclear how exactly losses and gains affect people's decisions.

Do these asymmetries between losses and gains extend to social situations, which often involve additional motivations such as fairness[14,15], reciprocity[16], and betrayal aversion[17]? Several studies compared social dilemmas with outcomes that were typically all positive or all negative. Several studies tested loss aversion (weighting losses stronger than gains), with mixed results: Some studies found evidence for loss aversion[18–20], but other studies did not, despite similar methods[21,22]. Some studies reported evidence for loss avoidance without disentangling loss avoidance from loss aversion[12,13,23,24]. Thus, in social dilemmas, the evidence for loss aversion is conflicting, and the evidence for loss avoidance is sparse. It is therefore unclear whether, and if so, how, losses and gains influence human decisions in social dilemmas.

In this article, we systematically varied losses and gains in social dilemmas and tested how this affected cooperation. Based on simulations, we identified experimental conditions that disentangled loss aversion and loss avoidance and allowed us to test both simultaneously. We ran seven behavioural studies in which participants decided in social dilemmas (Prisoner's Dilemma, Stag Hunt, and Chicken) with outcomes that could be gains, losses, or combinations thereof. Our overall predictions were that loss avoidance and loss aversion would influence people's cooperative decisions.

## Methods
### Experimental design
We tested for loss aversion and loss avoidance across seven behavioural experiments (see Table 1 for an overview of the differences between all experiments). Experiments 1–3 were conducted in the lab with iterated choices. Experiments 4–7 were one-shot games without feedback. Experiments 4 and 5 were pilot experiments that translated the approach from the iterated Prisoner's Dilemma games to one-shot Prisoner's Dilemma games in the lab (Experiment 4) and online (Experiment 5). Experiments 6 and 7 were preregistered replications of Experiment 5 that extended our results by finding evidence for loss avoidance in a one-shot Stag Hunt and Chicken game, where loss avoidance and loss aversion lead to opposite predictions, ensuring that loss avoidance was not specific to the Prisoner's Dilemma, and that people avoid losses, irrespective of whether this increased or decreased cooperation.

For the iterated experiments, we tested groups of six participants (who did not know each other) in the same room with six computers side-by-side, separated by partitions that prevented participants from seeing each other's screens. Participants played a series of iterated Prisoner's Dilemma games, in which we systematically varied whether the outcomes were positive or negative (see "Payoff matrices" below). In each group, each of the six participants played iterated Prisoner's Dilemmas with each of the other five participants, without knowing with whom they were playing. Each player

[1]Section Social Neuroscience, Department of General Psychiatry, Heidelberg University, Heidelberg, Germany. [2]Institute for Systems Neuroscience, University Medical Center Hamburg-Eppendorf, Hamburg, Germany. ✉e-mail: bjks.science@gmail.com; christoph.korn@med.uni-heidelberg.de

**Table 1 | An overview of the experimental designs for all experiments**

| Experiment | Where | Type | Game(s) | Decisions per participant | Incentive | Country | Preregistered |
|---|---|---|---|---|---|---|---|
| 1 | Laboratory | Iterated | Prisoner's dilemma | 300 | Yes | DE | - |
| 2 | Laboratory | Iterated | Prisoner's dilemma | 300 | Yes | DE | https://osf.io/dzkqg |
| 3 | Laboratory | Iterated | Prisoner's dilemma | 300 | Yes | DE | https://osf.io/z4geq |
| 4 | Laboratory | One-shot | Prisoner's dilemma | 1 | No | DE | - |
| 5 | Online (Prolific) | One-shot | Prisoner's dilemma | 1 | No | UK | - |
| 6 | Online (Prolific) | One-shot | Stag Hunt, prisoner's dilemma | 2 | No | UK | https://osf.io/72fj6 |
| 7 | Online (Prolific) | One-shot | Chicken, Stag hunt, Prisoner's dilemma | 3 | No | UK | https://osf.io/p6xdr |

In the 'country' column, DE = Germany and 'UK' = United Kingdom of Great Britain and Northern Ireland.

made 300 decisions: five (interactions) * 12 (payoff matrices) * five (repetitions). In Experiments 2 and 3, participants played three repetitions of 20 payoff matrices against each other player. The order of the payoff matrices within each interaction was randomised (but each repetition of a payoff matrix was played successively in each interaction). At the beginning of each interaction, both players were randomly assigned to be either the column-player (indicate decision with left-right arrow buttons) or the row-player (indicate decision with top-down arrow buttons) for the entire interaction. Participants were paid a base payment of 9€/h. We incentivized participants' decisions such that, depending on their decisions, up to 2€ (Experiment 1) or 5€ (Experiments 2 and 3) could be subtracted from or added to their base payment.

For the one-shot experiments, participants either participated in the lab (Experiment 4) or online (Experiments 5–7). For all one-shot experiments, participants made a single decision without feedback; the decisions were hypothetical and not incentivized. In Experiments 4–5, participants only made one decision (Prisoner's Dilemma), but in Experiment 6, participants decided first for Stag Hunt and then for the Prisoner's Dilemma, and in Experiment 7, participants decided first for Chicken, then for Stag Hunt, and then for the Prisoner's Dilemma. For Experiments 5–7, we also asked participants what they expected the other person to do, ranging from 0 (definitely defect) to 100 (definitely cooperate), but substituting the words "cooperate" and "defect" with the generic labels given to the response options. This question was presented after the experimental decisions.

We obtained informed consent to participate from all participants. The lab-based studies were approved by the ethics committee of the medical association (Ärztekammer) Hamburg (PV5746), and the online studies were approved by the ethics committee of the medical faculty at the University of Heidelberg (S-127/2020).

### Payoff matrices
We systematically varied whether the outcomes of the Prisoner's Dilemma games were positive or negative. We took a positive payoff matrix (e.g., in Experiment 1: $[T = 7, R = 5, P = 3, S = 2]$) and repeatedly subtracted a constant from all payoffs. We then categorized all payoff matrices into one of five payoff matrix categories: Category 1 (C1), in which all outcomes were positive; C2, in which all outcomes were positive apart from $S$, which was negative; C3, in which $T$ and $R$ were positive, and $P$ and $S$ were negative; C4, in which all outcomes were negative, apart from $T$, which was positive, and C5, in which all outcomes were negative. In Experiment 3 we also manipulated stake size, using four different levels, with a range of *8 of the smallest stake size. This range was limited by our incentivisation, which added/subtracted up to 5 Euros from the base payment. For all payoff matrices used in all experiments, see Table S1.

### Demographics
The participants for Experiments 1–3 were recruited through "Stellenwerk Hamburg", an internet-based platform for advertising small jobs. The experiment took part at the Institute for Systems Neuroscience at the University Medical Center Hamburg-Eppendorf.

The participants for Experiment 4 were recruited by asking participants taking part in other (non-social or decision-making) experiments at the Institute for Systems Neuroscience at the University Medical Center Hamburg-Eppendorf whether they had 5 min after their experiment ended to fill in a short questionnaire.

The participants for Experiments 5–7 were recruited via Prolific and tested on SoSciSurvey.de, a web platform for collecting online data. In Prolific, we only included participants from the UK who were native English speakers, with a prior approval rate of >95% (to pre-emptively exclude unserious participants); we pre-emptively excluded anyone who participated in any of our previous experiments, to remove the possibility of participants appearing in multiple datasets. We used Prolific's "balanced gender" feature, such that there were equal number of men and women in each sample. Slight variations in each experiment and condition can exist due to random allocations into groups, and due to failing comprehension tests.

For all Experiments, we asked participants to indicate their ages (in years). For Experiments 1–4, we asked participants to indicate their gender, offering "male" and "female" as options; for Experiments 5–7, we also added the options "diverse" and "prefer not to say." We did not collect any data on race or ethnicity.

For all experiments, we excluded participants (see Table 2) for two reasons: 1) if we thought that participants might not have understood the task properly, and 2) due to taking part in multiple experiments. These exclusions were always done in the same manner: in the iterated experiments, we defined a participant as not understanding the task properly if they failed to press the appropriate keys in the required time more than 10% of all trials, and in the one-shot experiments, it was defined by making even a single error in the test questions. We excluded such participants. Additionally, in the iterated experiments, due to administrative error, three participants (one in Experiment 2, two in Experiment 3) were excluded because they had taken part in one of the previous experiments. We excluded these participants because Experiments 2 and 3 were intended to be replications and extensions with independent data.

For Experiments 1–3, participants received detailed instructions with test questions, and experimenters answered any questions. Participants were excluded (see Table 2 for N of participants total and included in all experiments) if they gave too many (>10%) late/invalid responses or had taken part in one of the previous experiments.

For Experiments 4–7, which were very short (<5 min), we provided an explanation of the task and a few test questions. We excluded participants if they made any errors in the test questions (three test questions in Experiment 4, four test questions in Experiment 5 and two test questions in Experiments 6 and 7). Alternatively, we could have offered further explanation and test questions to participants who made an error in the test questions in the one-shot experiments, but this would have increased the average study duration. Instead, we opted for the financially more efficient route of excluding anyone who might not have understood the task.

For details for participants' demographics across all experiments, see Table 2.

#### Deriving hypotheses from simulations

To derive precise predictions for loss avoidance and loss aversion, we simulate people's decision-making with value functions that link objective outcomes to subjective value (how much people value each outcome). Loss avoidance has been formalized by an intercept of 0 for gains and a negative intercept for losses[13]. In contrast, loss aversion is typically formalized by a steeper slope for losses than gains[9] (Fig. 1). We use these simulations to illustrate the categorical predictions of loss avoidance and loss aversion if people acted according to simple decision models. In particular, the simulations provide an intuitive yet theoretically based understanding of how loss avoidance would affect cooperation in the various games.

We model social interactions with social dilemmas, specifically 2 × 2 games. In 2 × 2 games, two players decide between two actions (Fig. 2a), leading to four possible outcomes for each player. Depending on the ordering of the outcomes, different game-theoretic situations emerge, such as the Prisoner's Dilemma, Chicken, or Stag Hunt (Fig. 2b–d). We shift all payoffs of a 2 × 2 game for losses and gains by subtracting a constant (Fig. 2e, f). The resulting games can be categorized into five distinct categories: in Category 1, all outcomes are positive; in Category 2, all outcomes are positive, apart from the lowest, which is negative; In Category 3, the two highest payoffs are positive and the two lowest payoffs are negative; in Category 4, all outcomes are negative apart from the highest, which is positive; in Category 5, all outcomes are negative. In the following, we will use these five categories to test both loss aversion and loss avoidance at the same time.

The basic idea for simulating people's decisions is that people choose the option with the higher subjective utility. The utility of each option is given by a linear formula for expected utility: u(x) = px, where x is the payoff and p is the probability of it occurring. For C and D, we add the utilities of the two possible outcomes in a 2 × 2 game. Therefore, the expected utilities are given by

$$u(C) = pR + (1 - p)S \qquad (1)$$

$$u(D) = pT + (1 - p)P \qquad (2)$$

where p is the probability that the other person will cooperate, and {T, R, P, S} are the four possible outcomes in a 2 × 2 game.

To account for potential asymmetries between payoffs in the gain and loss domains, we filtered each outcome {T, R, P, S} through a simple function that differentiated the slope between losses and gains and added an intercept for losses:

$$y = \begin{cases} x, & y \geq 0 \\ \lambda x - b, & y < 0 \end{cases} \qquad (3)$$

We use a sigmoid function to model people's responses when choosing between C and D. This leads to qualitative predictions for the average cooperation rate in each of the five categories for each social dilemma (see Fig. 3). Specifically, for the Prisoner's Dilemma, a player with loss avoidance would have a lower cooperation rate for Categories 2 and 4 compared to Categories 1, 3, and 5, because in Categories 2 and 4, where only one outcome is negative (Category 2) or only one outcome is positive (Category 4), defection can guarantee that one does not lose (Category 2) or be the only way in which one can avoid a loss (Category 4, if the other cooperates); similarly, a player with loss aversion would show a steady decline in cooperation rate from Categories 1 to 5.

We use two different models to compare loss avoidance (same slope with an intercept) and loss aversion (steeper slope in the loss domain, with no intercept). For loss aversion, we chose $\lambda = 2$, which was commonly thought to be a representative value in the population (but that recent meta-analyses suggest are overestimated[25,26]. Using lower values for loss aversion would lead to the same pattern, but with a smaller effect size). For loss avoidance, we chose $b = 4$. The magnitude of the parameter mainly influences the magnitude of the effect (e.g., a larger loss avoidance parameter would lead to a stronger reduction of cooperation for Categories 2 and 4 in the Prisoner's Dilemma; a larger loss aversion parameter would lead to a steeper downward shift from Categories 1 to 5 in the Prisoner's Dilemma). For extreme loss aversion, the patterns break down (e.g., if loss aversion were extremely large, cooperation would be 0% for any payoff matrix with a loss), but within normal ranges, the size of the parameters merely manipulates the magnitude of the effect. A player can also have both influences, which would be a linear combination of the two individual predictions (i.e., a steady

### Table 2 | Demographics for all experiments

| Experiment | N included (N total) | Age mean (SD) | Gender % male/female/ diverse/prefer not to say |
|---|---|---|---|
| 1 | 24 (24) | 28 (8) | 42/58/NA/NA |
| 2 | 46 (48) | 27 (5) | 39/61/NA/NA |
| 3 | 44 (48) | 27 (7) | 41/59/NA/NA |
| 4 | 83 (93) | 25 (5) | 44/56/NA/NA |
| 5 | 447 (902) | 39 (12) | 51/47/2/1 |
| 6 | 603 (751) | 41 (13) | 51/47/1/1 |
| 7 | 603 (754) | 40 (13) | 50/49/1/0 |

Experiments 1–4 include 'NA' in the gender column because we only included 'female' and 'male' as response options.

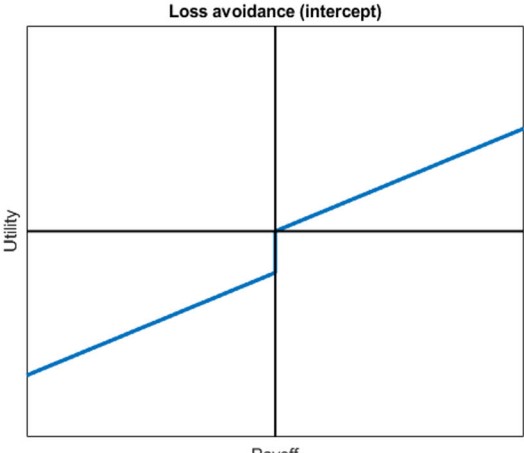

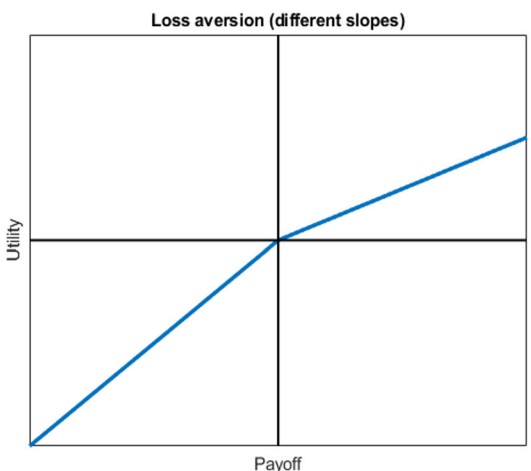

**Fig. 1 | Linear utility curves for loss avoidance (left) and loss aversion (right).** We formalize loss avoidance with an intercept of 0 for gains and a negative intercept for losses, and loss aversion with a steeper slope for losses than for gains.

|   | C | D |
|---|---|---|
| C | R/R | S/T |
| D | T/S | P/P |

(a) Generic form

|   | C | D |
|---|---|---|
| C | 5/5 | 1/7 |
| D | 7/1 | 3/3 |

(b) Pris. Dilemma

|   | C | D |
|---|---|---|
| C | 5/5 | 3/7 |
| D | 7/3 | 1/1 |

(c) Chicken

|   | C | D |
|---|---|---|
| C | 7/7 | 1/5 |
| D | 5/1 | 3/3 |

(d) Stag-Hunt

Category 1:   $T > R > P > S > 0$
Category 2:   $T > R > P > 0 > S$
Category 3:   $T > R > 0 > P > S$
Category 4:   $T > 0 > R > P > S$
Category 5:   $0 > T > R > P > S$

(e)

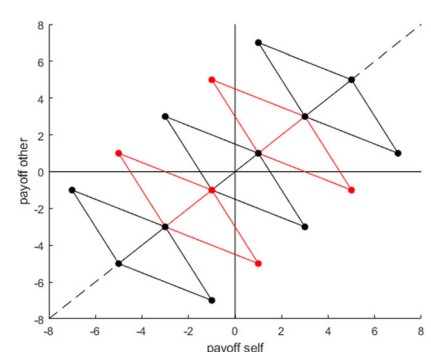

(f)

**Fig. 2 | - 2×2 games and the five categories of payoff matrices.** 2 players each decide between 2 options (**C** or **D**), leading to four possible outcomes per person **a** Depending on the ordering of the outcomes, different situations emerge, such as **b** the Prisoner's Dilemma ($T > R > P > S$), **c** Chicken ($T > R > S > P$), or **d** Stag Hunt ($R > T > P > S$). **e** The inequalities display the five possible categories of payoff matrices for a Prisoner's Dilemma if outcomes can be gains or losses. For other games, the principle for the five categories is the same, but the order of the payoffs is different (Stag Hunt: $R > T > P > S$; Chicken: $T > R > S > P$); **f** provides a visual representation of (**e**) of the payoffs for both players for each of the five categories.

decline in cooperation rate from Categories 1 to 5, with specific dips for Categories 2 and 4).

So far, we have only simulated loss aversion and loss avoidance in the Prisoner's Dilemma. But the Prisoner's Dilemma is only one social dilemma, where loss avoidance and loss aversion always reduce cooperation. Ideally, we would also test our hypotheses in social dilemmas where loss aversion and loss avoidance can each increase and decrease cooperation in different situations. Two ideal games are Stag Hunt ($R > T > P > S$) and Chicken ($T > R > S > P$), which are identical to the Prisoner's Dilemma, each with one exception: For Stag Hunt, the two highest payoffs (T and R) are swapped, such that in the Prisoner's Dilemma defecting on a cooperator is the best outcome, but in Stag Hunt mutual cooperation is the best outcome. This swap changes the simulations' predictions for loss avoidance and loss aversion (see Fig. 3 top/blue and middle/orange), such that both loss avoidance and loss aversion sometimes increase cooperation and some decrease cooperation. For Chicken, the two lowest outcomes are swapped relative to the Prisoner's Dilemma, which leads to analogous changes in the predictions (see Fig. 3, bottom/red). Finding evidence for loss avoidance in all three games with the same participants would provide robust evidence that this is a general finding in social dilemmas and not due to some quirk of the Prisoner's Dilemma.

We simplify our simulations with two common assumptions[6,27]:

First, we assume linear utility functions rather than the convex/concave utility functions specified by Kahneman & Tversky[9]. Fox and Poldrack[28] reviewed studies that estimated the parameters for Prospect Theory. For the studies that estimated exponential parameters for losses and gains using the same functional form we use, these functions are practically linear for our purposes (the median exponential parameter was 0.88 for gains and 0.92 for losses, which results in very similar predictions to the simplified linear model). Using non-linear exponents sometimes found in empirical studies (e.g., an exponent of 0.7) can lead to similar predictions to those made by loss avoidance. Thus, if participants were to have extreme non-linear exponents for the ranges we use in our experiments, part of our effect might stem from diminishing returns rather than from loss avoidance. Additionally, three slightly different definitions of loss aversion coincide under such a piecewise linear simplification.

Second, we assumed that players expect a 50:50 chance that the other player will cooperate or defect (as a control, we later inserted participants' estimated probability of what the other person would do, which led to indistinguishable predictions). Our simulated predictions require mixed outcomes and therefore do not hold at the extreme ends of p (i.e., near 0 and 1): if a player knows that the other will cooperate or defect, many gain/loss asymmetries no longer apply. For example, if a player in the Prisoner's Dilemma Category 2 knows with certainty that the other person will cooperate, then the only loss outcome of the payoff matrix (player one cooperates and player two defects) is not possible for player one. In other words, our hypotheses require the possibility of losses and gains for asymmetries between the two to become relevant for people's decisions (see Fig. S3).

Further, we assumed that participants would have a roughly 50% cooperation rate in the "neutral" (C1, all outcomes are positive) condition in the Prisoner's Dilemma, based on similar previous experiments (and in our experiments, C1 had a roughly 50% cooperation rate in the Prisoner's Dilemma). Formally, to define a 50% cooperation rate for the Prisoner's Dilemma in our simulations, we set the inflection point of the sigmoid function (with beta = 1) to be the difference in expected utility between the D and the C option. As the difference in expected utility is neutral for Stag Hunt and Chicken, this led to a predicted higher overall cooperation rate, which was also borne out by the empirical data. Note that the precise cooperation rate in the neutral condition is not essential for our categorical predictions, as long as it allows for variation (e.g., if the neutral categories had a cooperation rate of 0% in the Prisoner's Dilemma, loss avoidance could not reduce that any further).

## Statistical analysis
For all statistical analyses, see the openly shared code.

Prisoner's Dilemma: For the iterated experiments, we calculate the average cooperation rate for each participant in each of the five Categories. To test loss aversion, we used a repeated-measures $t$ test to compare Categories 1 and 5 (more cooperation in Category 1 compared to Category 5 indicating loss aversion). To test loss avoidance, we further calculated the average cooperation rate per participant for Categories 2 and 4, and for

**Fig. 3 | The simulated predictions for each game (left column) and a summary of the (empirical) cooperation rates across all experiments for the five categories (right column).** Based on simulations, we predicted cooperation rates for each game for each category, assuming an agent has a utility curve with loss avoidance (top) or loss aversion (bottom). For details of the simulations, see the first section in Results, Fig. 2 and Methods. Results: Blue top: iterated Prisoner's Dilemma; blue bottom: one-shot Prisoner's Dilemma; orange: one-shot Stag Hunt; red: one-shot Chicken. Experiments 1–5 used the Prisoner's Dilemma, Experiment 6 used Stag Hunt and the Prisoner's Dilemma, and Experiment 7 used Chicken, Stag Hunt, and the Prisoner's Dilemma. Experiments 1–3 were iterated and Experiments 4–7 were one-shot. Experiments 1–4 were offline in the lab, Experiments 5–7 were online. For illustrative purposes, we combined the data from similar experiments here (for the iterated Experiments 1–3, we took the mean cooperation rate of each participant in each category, combined them, and calculated a mean of means for each Category; for the one-shot Experiments 4–7, we combined all (binary) choices for each category for each game, and calculated a new mean response), even though almost all statistical tests were done separately for each experiment. Experiments 1–3 are displayed with the median cooperation rate per category, plus its 95% confidence interval, the 25th and 75th percentile, and the range. For the one-shot games, where each participant made a single decision per game, there were no individual means, so we display the group means; error bars indicate the standard error of the mean; black dots indicate individual decisions. The numbers (in parentheses) on the x-axis indicate the number of participants included in each bar.

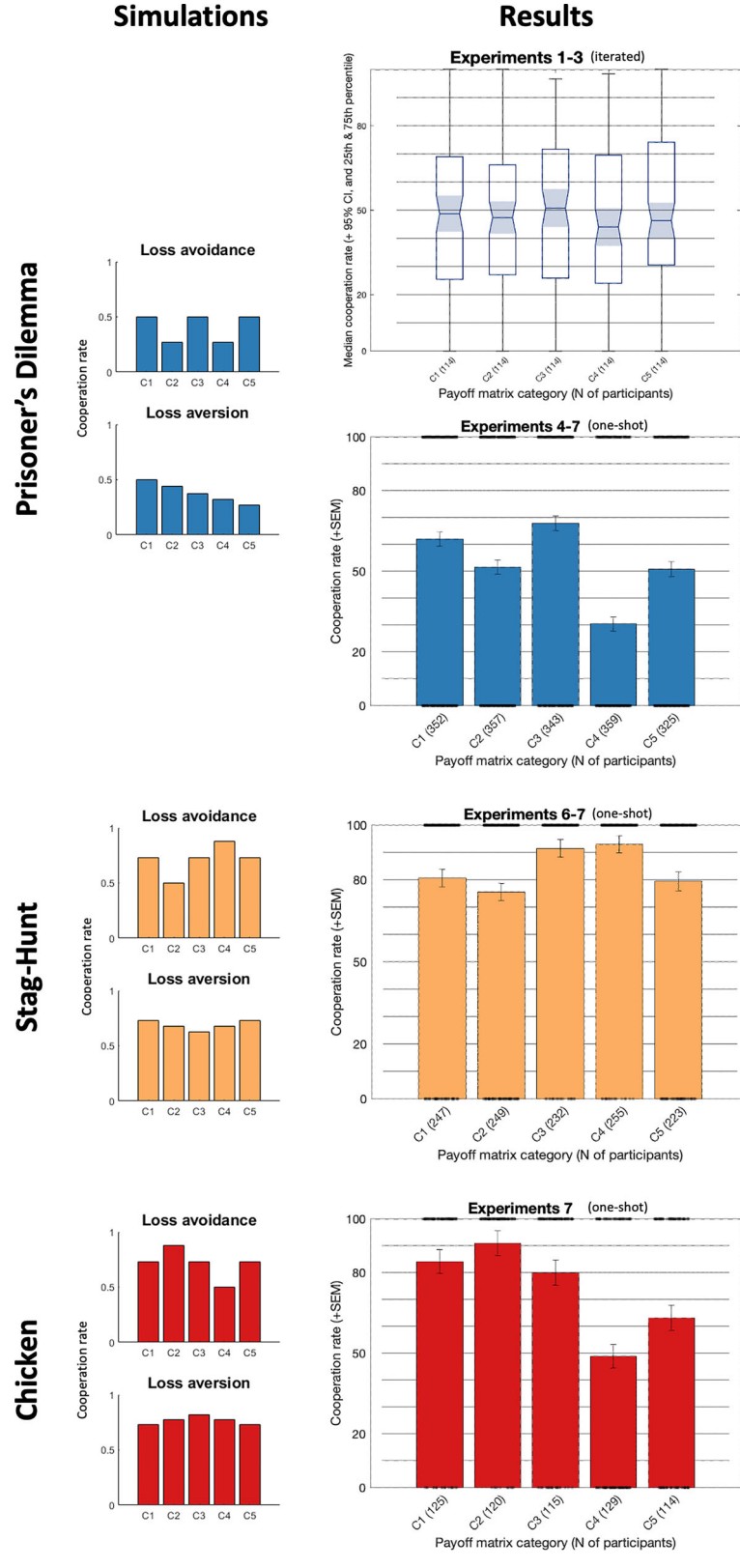

Categories 1, 3, and 5 (more cooperation in Categories 1, 3, and 5, compared to Categories 2 and 4, indicating loss avoidance). Experiments 1 and 2 used a repeated-measures t-test to compare their cooperation rate. For Experiment 3, we used a repeated measures ANOVA to test for a main effect of stake size and its interaction with loss avoidance. We added an exploratory logistic mixed effects model for the iterated experiments with Participant in Group

in Experiment as nested random effect and loss aversion as an ordered predictor. In the one-shot Experiments 4-7, participants make a single decision (per game), so calculating cooperation rates per participant is impossible. Therefore, we used Chi-squared tests to run analogous tests for the one-shot experiments. Similarly, we also ran logistic mixed effects models over the whole group (without nested effects).

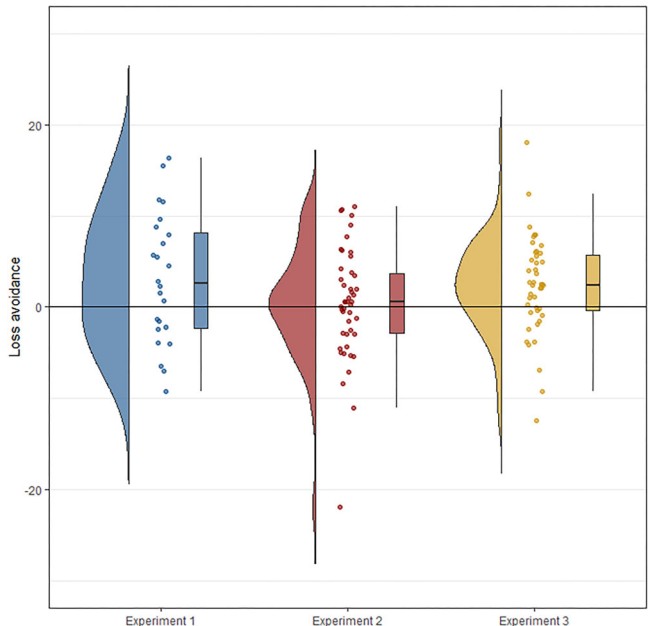

**Fig. 4 | Loss avoidance in the iterated Experiments 1–3.** For each experiment, the figure displays raincloud plots for loss avoidance, consisting of probability distribution, individual data points, and box plots (median, interquartile range, and 1.5 * interquartile range). Loss avoidance is calculated by taking the difference per participant between their cooperation rate in Category(1, 3, 5) - Category(2, 4), such that a positive score indicates loss avoidance. N of participant Experiment 1: 24; Experiment 2: 46; Experiment 3: 44.

**Table 3 | Deviations from the preregistered plan**

| Preregistration | Deviations |
|---|---|
| Experiment 2 | Preprocessing: we followed the preregistration precisely apart from one exception: in addition to the preregistered exclusion criteria (1: trials with invalid response, 2: trials with late/no response, and 3: participants with more than 10% of invalid/late/no responses), we excluded one participant after realizing that that participant had already taken part in Experiment 1. This additional exclusion criterion was not preregistered for Experiment 2, but was included for the preregistration of Experiment 3 |
| Experiment 2 | Analysis: We preregistered additional analyses to further characterize the significant main effects we expected to find for loss avoidance and loss aversion. The main effects were not significant, so we did not run these preregistered follow-up analyses. |
| Experiments 2 & 3 | Change of terminology: We changed the naming of the categories of payoff matrices from the descriptive ALLPos/Spec1/Midd/Spec2/AllNeg to the more neutral categories 1–5 used in the manuscript. Similarly, in the initial preregistrations we used the terms 'loss aversion' incorrectly: we mentioned ''loss aversion'' but meant ''loss avoidance'' as defined by an intercept, rather than a differences in slopes. This was merely an error of naming, the logic of testing the effect remains the same and we didn't make any changes from the preregistered analysis plan (once you change the names). |
| Experiments 6 & 7 | No deviations from preregistered plan |

Stag Hunt and Chicken: The analyses for Stag Hunt and Chicken are identical to the Prisoner's Dilemma, with the sole exception that the predicted cooperation rates differ for the different Categories, such that the comparisons need to be adjusted: First, in the Prisoner's Dilemma we could test loss avoidance by combining the data from Categories 2 and 4. This was possible because the predictions were identical for those two Categories for loss avoidance and the Prisoner's Dilemma. However, Categories 2 and 4 lead to opposite predictions for both Stag Hunt and Chicken, so they cannot be combined. Instead, in Stag Hunt, we test loss avoidance by comparing Category 4 to Categories 1, 3, and 5, and to Category 2, and by comparing Categories 1, 3, and 5 to Category 2 (Category 4 should have the highest cooperation rate, followed by Categories 1, 3, and 5, and with Category 2 with the lowest cooperation). For Chicken, we run the same analyses but with opposite predictions: Category 2 should have the highest cooperation rate, followed by Categories 1, 3, and 5, with Category 4 having the lowest cooperation rate. For loss aversion, Stag Hunt should have the lowest cooperation rate in Category 3 and the highest cooperation rates in Categories 1 and 5, so we again run a Chi-squared test by comparing Categories 1 and 5 with Category 3. Again, the same is true for Chicken, but with opposite predictions. Here, too, we run logistic mixed-effects models at the group level.

For the logistic regressions in Experiments 6 and 7, we preregistered to also include the expectation of what the other person would do as a predictor in the model, both as main effect, and as interaction effect with loss avoidance and loss aversion. We made the conceptual error that expectation is a post-treatment variable, which makes its addition to the logistic regressions difficult to interpret. As preregistered analyses, we still include the full results, but we prioritise the Chi-squared tests, as these 1) do not suffer from the same problem, and 2) align with the logistic regressions (setting aside the expectation effects).

The assumptions of all tests were met: Iterated analyses: for the repeated $t$ tests and the repeated-measures ANOVA, we calculated average cooperation rates across all trials of a given contrast for each participant, leading to continuous data that is approximately normally distributed (see also Fig. 4; there is one large outlier in Experiment 2 for loss avoidance, but excluding this participant does not change the results in any meaningful way). For the exploratory logistic mixed effects models, the data from participants is not independent, as they interact in real time with immediate feedback; we therefore ran these models with nested effects, where each participant takes part in a given group, which is part of a given experiment. We chose a logistic regression because each decision was binary, and the predictors of loss aversion and loss avoidance were uncorrelated. One-shot

analyses: for the Chi-squared tests, the data was binary, consisted of independent observations (one per participant for each test), and had large cell counts; for the logistic regressions, the outcomes were binary, each participant made only one decision per game, we had a large sample size, and loss aversion and loss avoidance were uncorrelated predictors. The only concern was about the predictor "expectation of what other will do," which was a post-treatment variable, which depends on the condition and correlates strongly with cooperation (as explained above).

### Deviations from preregistrations
Four experiments were fully preregistered: Experiment 2: https://osf.io/dzkqg (18.07.2019); Experiment 3: https://osf.io/z4geq (01.09.2019); Experiment 6: https://osf.io/72fj6 (05.09.2022); Experiment 7: https://osf.io/p6xdr (06.09.2022).

We made minor changes to the preregistered plan for Experiments 2 and 3; Experiments 6 and 7 followed the preregistrations exactly (see Table 3).

## Results
### Weak evidence from iterated Prisoner's Dilemmas (Experiments 1-3)
We first tested the effects of loss aversion and loss avoidance on social dilemmas by running three laboratory experiments (Ns = 24, 46, 44) in which participants played iterated Prisoner's Dilemmas in groups of 6. Each participant played various payoff matrices split evenly across the five categories, iteratively for 60 rounds with each of the five other players. There was no deception; participants received real-time feedback about their partners' choices. We incentivized participants' decisions, such that their decisions in the experiment were converted into real currency, which was added to or subtracted from their base payment. The first experiment was a pilot study, and the other two were preregistered replications.

The overall evidence from the three experiments suggested a weak effect of loss avoidance but there was no evidence for loss aversion (see Fig. 3 blue, top for a summary of all three experiments; see Fig. 4 for loss avoidance of each participant): in the initial pilot Experiment 1, the mean cooperation rate per participant for Categories 2 and 4 combined (M = 47.2%, SD = 24.7) was significantly lower than the mean cooperation rate of Categories 1, 3, and 5 combined (M = 50.3%, SD = 24.3; t(23) = 2.1, p = 0.045, d = 0.126, 95% CI [−6.1, −0.1]), providing evidence for loss avoidance. The cooperation rate of Category 1 (M = 47.3%, SD = 24.7) was significantly lower than that of Category 5 (M = 54.2%, SD = 24.1; t(23) = -2.6, p = 0.016, d = 0.284, 95% CI [−12.4, −1.4]), indicating the opposite of loss aversion. We then ran a preregistered replication (Experiment 2) but found neither of our previous findings supported: The same analysis used in Experiment 1 revealed no significant effects for loss avoidance or loss aversion: The mean cooperation rates between Categories 2 and 4 combined (M = 46.0%, SD = 32.2) and Categories 1, 3, and 5 combined (M = 46.0%, SD = 30.8) did not differ significantly (t(45) = -0.4, p = 0.687, d = 0.012, 95% CI [−2.2, 1.5], $BF_{10}$ = 0.17), nor were there significant differences between the cooperation rates of Category 1 (M = 45.9%, SD = 31.0) and Category 5 (M = 45.7%, SD = 31.2; t(45) = 0.2, p = 0.883, d = 0.007, 95% CI [−2.7, 3.1], $BF_{10}$ = 0.16). Experiments 1 and 2, therefore, find conflicting results. We thus preregistered **Experiment 3** to clarify this discrepancy and to test for potential interaction effects of stake size and loss aversion[29,30]. There was a significant effect of loss avoidance on cooperation (F(1, 43) = 8.37, p = 0.006), but neither stake size (F(3, 43) = 0.99, p = 0.400, $BF_{10}$ = 0.67) nor the interaction between loss avoidance and stake size (F(3,43) = 1.07, p = 0.363, $BF_{10}$ = 0.32; see Fig. S1) had a significant effect on cooperation. We again found no evidence for loss aversion: the cooperation rates of Category 1 (M = 50.0%, SD = 26.2) and Category 5 (M = 48.8%, SD = 25.2) did not differ significantly (t(44) = 0.9, p = 0.362, $BF_{10}$ = 0.25). For Experiments 2 and 3, we also measured Social-Value Orientation[31] and predicted it to correlate with loss avoidance, which was not the case in either experiment (see Fig. S2).

The results from Experiments 1–3 are thus mixed: Experiments 1 and 3 find evidence for loss avoidance but Experiment 2 does not; Experiment 1

finds evidence for the opposite effect of loss aversion, but neither Experiment 2 nor 3 finds any significant effect of loss aversion. The methodological set-up of Experiments 1–3 was practically identical, so we combined the data to run more comprehensive logistic mixed effects models on the combined dataset from all 114 participants. This analysis was exploratory and not preregistered. We aimed to 1) clarify the role of the null results for loss avoidance from Experiment 2, 2) run hierarchical models by using nested random effects of participant in group in experiment; this corrects for correlated standard errors and explicitly models the nested structure of the experimental design (which the previous analyses do not), 3) test loss aversion in a more nuanced way: our simulations predicted a linear decline in cooperation from Category 1 to Category 5 for loss aversion, so we added Category as an ordered predictor, rather than contrasting Category 1 with Category 5, and 4) ensure that any results for loss avoidance or loss aversion are not confounded by the interactive history between each pair of players: Based on the extensive literature on reciprocity[32], we expected that the interaction history would have a large effect on people's decisions. Indeed, across Experiments 1–3, 79% of all decisions were the same as the other person's previous move (generally aligned with the strategy Tit-for-Tat). We therefore also included the other person's previous decision as a predictor to ensure that any effects of loss avoidance or loss aversion were not due to the interaction history.

Combining the data from Experiments 1–3 into a logistic mixed effects model with Participant in Group in Experiment as nested random effect, we found a significant effect of loss avoidance on cooperation rate (β = -0.151, SE = 0.035, 97.5% CI = [−0.219, −0.083], z = -4.358, p < 0.001), with no evidence for an effect of loss aversion (β = 0.038, SE = 0.034, 97.5% CI = [−0.029, −0.105], z = 1.120, p = 0.263); the other person's previous decision was a very strong predictor (β = 2.495, SE = 0.032, 97.5% CI = [2.433, 2.557], z = 78.646, p < 0.001; see Table S2 for full table).

In summary, our exploratory logistic mixed-effects models using the pooled data from Experiments 1-3 suggest that: 1) loss avoidance significantly reduced cooperation rate; this effect remains stable when adding loss aversion and interaction history as predictors, 2) there was no evidence that loss aversion affected cooperation, and 3) the interactive history between two players had a substantial effect on people's subsequent behaviour.

### Strong evidence from one-shot prisoner's dilemmas (Experiments 4–7)
Given the large effect of interaction history, we reasoned that loss avoidance might be larger in one-shot interactions, where interactive effects cannot obscure the effect of loss avoidance (e.g., when playing with someone who always defects, most people would also defect, thus leaving little to no variance in the data). We therefore attempted to replicate and extend our findings from the iterated Prisoner's Dilemmas with one-shot Prisoner's Dilemmas. We repeated the logic from Experiments 1–3 without iterations: each participant was randomly assigned to one of the five categories of payoff matrices and made a single experimental decision (without feedback), leading to a pure one-shot game. Experiment 4 was in the lab but was interrupted by the COVID-19 pandemic, so Experiments 5–7 were online. Experiment 4 (N = 83) was the laboratory pilot, Experiment 5 (N = 447) was the online pilot, and Experiments 6 (N = 603) and 7 (N = 603) were preregistered replications and extensions. For practical reasons, the one-shot experiments were hypothetical and not incentivized. Although incentivized decisions would have been ideal, we do not believe this weakens our results. If people try to avoid imaginary financial losses, it seems likely that they would also try to avoid actual financial losses.

In the one-shot Prisoner's Dilemma games, there was consistent evidence with larger effect sizes for loss avoidance and some evidence for loss aversion (see Fig. 3, blue, right bottom). For Experiment 4, the limited sample size (due to COVID-19) allowed us to test for loss avoidance only. Applying the same logic as in Experiments 1–3 (with only a single binary decision per participant), Categories 2 and 4 had significantly lower cooperation rates (38.7%) than Categories 1, 3, and 5 (63.5%), providing evidence

for loss avoidance ($\chi^2(1, N = 83) = 4.79$, $p = 0.029$, $\varphi = 0.24$, 95% CI [0.036, 0.444]). Given the limitation of the sample size of Experiment 4 and the ease of collecting large samples online, we conducted an online version of Experiment 4: In Experiment 5, the cooperation rate for Categories 2 and 4 (36%) was significantly lower than that of Categories 1, 3, and 5 (49%) ($\chi^2(1, N = 447) = 8.45$, $p = 0.004$, $\varphi = 0.14$, 95% CI [0.049, 0.231]), again providing evidence for loss avoidance. There was no significant difference in cooperation rate between Category 1 (51%) and Category 5 (43%) ($\chi^2(1, N = 175) = 1.00$, $p = 0.328$, $\varphi = 0.08$, 95% CI [−0.068, 0.228]; $BF_{10} = 0.13$), providing no evidence for loss aversion. We found further evidence for loss avoidance in Experiments 6 and 7. These experiments were primarily designed to extend our findings beyond the Prisoner's Dilemma (see below) but also served to replicate the findings from the one-shot Prisoner's Dilemma. In Experiment 6, Categories 2 and 4 (39%) had a significantly lower cooperation rate than Categories 1, 3, and 5 (66%) ($\chi^2(1, N = 603) = 42.25$, $p < 0.001$, $\varphi = 0.27$, 95% CI [0.196, 0.344]), providing evidence for loss avoidance. The difference between cooperation rates in Category 1 (68%) and Category 5 (54%) was statistically significant ($\chi^2(1, N = 231) = 4.70$, $p = 0.030$, $\varphi = 0.14$, 95% CI [0.013, 0.267]), providing evidence for loss aversion. Experiment 7 closely replicated Experiment 6: There was evidence for loss avoidance: the average cooperation rate of Categories 1, 3, and 5 (63%) was again significantly higher than the cooperation rate of Categories 2 and 4 (47%; $\chi^2(1, N = 603) = 15.03$, $p < 0.001$, $\varphi = 0.16$, 95% CI [0.082, 0.238]). The difference in cooperation rates between Category 1 (66%) and Category 5 (52%) was also statistically significant ($\chi^2(1, N = 239) = 4.73$, $p = 0.030$, $\varphi = 0.14$, 95% CI [0.015, 0.265]), suggesting that participants were loss averse. Thus, across four one-shot Prisoner's Dilemma experiments, we find very consistent evidence for loss avoidance and some evidence for loss aversion.

### Strong evidence from one-shot stag hunt and chicken games

Experiments 6 and 7, mentioned above in the context of replicating the Prisoner's Dilemma, were designed primarily to extend our results from the Prisoner's Dilemma to other contexts. While Experiment 6 added a one-shot Stag Hunt game, Experiment 7 added one-shot Stag Hunt and Chicken games to provide an independent replication of the Stag Hunt results from Experiment 6, too.

**Experiment 6**. For Stag Hunt, we predicted finding loss avoidance, which meant that Category 4 would have a higher cooperation rate than Categories 1, 3, and 5, which should have a higher cooperation rate than Category 2 (and therefore, Category 4 should have a higher cooperation rate than Category 2). All three hypotheses for loss avoidance were confirmed: Category 4 (99%) had a higher cooperation rate than Category 2 (77%) ($\chi^2(1, N = 255) = 30.12$, $p < 0.001$, $\varphi = 0.34$, 95% CI [0.231, 0.449]) and Categories 1, 3, and 5 (86%) ($\chi^2(1, N = 474) = 17.75$, $p < 0.001$, $\varphi = 0.19$, 95% CI [0.103, 0.277]), which in turn had a higher cooperation rate than Category 2 ($\chi^2(1, N = 477) = 5.33$, $p = 0.021$, $\varphi = 0.11$, 95% CI [0.021, 0.199]). A lower cooperation rate in Category 3 than in Categories 1 and 5 would imply loss aversion. We found, however, that Category 3 had a significantly higher cooperation rate (92%) than Categories 1 and 5 (82%) ($\chi^2(1, N = 348) = 6.38$, $p = 0.012$, $\varphi = 0.14$, 95% CI [0.037, 0.243]), the opposite of loss aversion. To run more nuanced tests of loss avoidance and loss aversion, we also ran (preregistered) logistic regressions, in which we predicted cooperation rate from loss avoidance, loss aversion, and the expectation of what the other will do (plus interactions) separately for the Prisoner's Dilemma and for Stag Hunt (see Tables S3–8 for full tables for the logistic regressions for all one-shot experiments). The results from the logistic regressions are in line with the results for both games for loss avoidance, but when including expectation as a predictor, loss aversion in Stag Hunt was no longer a significant predictor. Thus, there was strong evidence for loss avoidance but mixed evidence for loss aversion, replicating our previous findings. Crucially, for Category 4, the Prisoner's Dilemma and Stag Hunt lead to opposite predictions for loss avoidance. In both games, this was confirmed: in Category 4, almost all

participants cooperated (99%) in Stag Hunt, but the same people mainly defected in the Prisoner's Dilemma (25%). Loss avoidance thus led to switching for most participants in Category 4 but much less so in the other categories. In summary, our findings extend to Stag Hunt: there was clear evidence for loss avoidance but mixed evidence for loss aversion.

**Experiment 7**. For Chicken, as predicted, Category 2 (91%) had a higher cooperation rate than Category 4 (49%) ($\chi^2(1, 478 N = 249) = 51.33$, $p < 0.001$, $\varphi = 0.45$, 95% CI [0.378, 0.522]) and Categories 1, 3, and 5 (76%) ($\chi^2(1, N = 474) = 12.23$, $p < 0.001$, $\varphi = 0.16$, 95% CI [0.072, 0.248]), which in turn had a higher cooperation rate than Category 4 ($\chi^2(1, N = 483) = 32.44$, $p < 0.001$, $\varphi = 0.26$, 95% CI [0.177, 0.343]), providing evidence for loss avoidance. For loss aversion, the difference in cooperation rate between Category 3 (80%) and Categories 1 and 5 (74%) was not statistically significant ($\chi^2(1, N = 354) = 1.50$, $p = 0.220$, $\varphi = 0.07$, 95% CI [−0.034, 0.174]; $BF_{10} = 0.12$). For Stag Hunt, our results from Experiment 6 replicated but were less clear: while Category 4 (87%) still had a significantly higher cooperation rate than Category 2 (74%) ($\chi^2(1, N = 249) = 6.44$, $p = 0.011$, $\varphi = 0.16$, 95% CI [0.039, 0.281]), it did not have a statistically significantly higher cooperation rate than Categories 1, 3, and 5 (82%) ($\chi^2(1, N = 483) = 1.63$, $p = 0.202$, $\varphi = 0.06$, 95% CI [−0.029, 0.149]; $BF_{10} = 0.11$), which in turn did not have a higher cooperation rate than Category 2 ($\chi^2(1, N = 474) = 3.36$, $p = 0.067$, $\varphi = 0.08$, 95% CI [−0.010, 0.170]; $BF_{10} = 0.23$). Category 3 again had a higher cooperation rate (90%) than Categories 1 and 5 (78%) ($\chi^2(1, N = 354) = 8.34$, $p = 0.004$, $\varphi = 0.15$, 95% CI [0.048, 0.252]), the opposite of what loss aversion would predict. As with Experiment 6, we also ran (preregistered) logistic regressions for each game (loss avoidance was a clear significant predictor in all three games; see S6–S8). As with Experiment 6, finding evidence for loss avoidance in all three games means that, to avoid losing money, many participants either cooperated more or cooperated less, depending on the combination of game and Category. In summary, our results extended to Chicken: there was clear evidence for loss avoidance but mixed evidence for loss aversion.

## Discussion

Across seven experiments, we systematically varied whether the outcomes of a Prisoner's Dilemma, Stag Hunt, and Chicken games were gains or losses. This allowed us to simultaneously test loss avoidance (utility function with intercept for losses) and loss aversion (different slopes). We found consistent evidence for loss avoidance across almost all experiments and tests. For loss aversion, the results were consistently inconsistent: there was as much evidence for loss aversion as for its opposite, and most tests showed no significant differences

Participants tried to avoid losses in all three social dilemmas studied, suggesting that the results are not due to some quirk of the Prisoner's Dilemma but apply to the most studied social dilemmas, independent of whether loss avoidance increases or decreases cooperation. The precise consequence of loss avoidance is thus determined by the context of the game and how this situation allows people to avoid losses. Participants tried to avoid a loss at the cost of the other person suffering such a loss: in Category 4 of the Prisoner's Dilemma and of Chicken, defecting (to avoid a loss if the other cooperates) confines the other player to a loss. For our participants, forcing a loss on the other person to prevent a loss for oneself was a price worth paying.

Our results largely match previous studies using 2 × 2 games, although these did not use all five categories: for loss avoidance, the studies closest to ours[12,13] investigated loss avoidance in the Stag Hunt game and used payoff matrices that we would classify as Categories 1, 2, and 4. While these studies provide evidence for loss avoidance, other factors might explain the results, such as different average payoffs in the categories or different levels of loss aversion. In our experiments, the comparisons of Categories 2 and 4 with Categories 1, 3, and 5 for the Prisoner's Dilemma ensure that each side of this comparison has the same average payoff and the same predicted level of loss aversion (and for Stag Hunt and Chicken the asymmetries are balanced

across all tests). Previous studies have tested loss aversion in the Prisoner's Dilemma by comparing Categories 1 with 5. Our mixed results match the results of earlier studies: some find no overall effects of loss aversion[21,22], but others do find such an effect[19,20]. Our one-shot results from the Prisoner's Dilemma largely replicate the effect of loss aversion reported by Sun et al. Still, the results from the same participants from the Stag Hunt and Chicken games in Experiments 6–7 do not find such evidence (or even the opposite). The iterated Prisoner's Dilemmas results also do not find consistent evidence for loss aversion. Our mixed results from loss aversion match the mixed results about loss aversion from non-social contexts[25,26,33–36]. The field might face a similar situation to the risk elicitation puzzle[37], where different ways of measuring the supposedly same phenomenon lead to different outcomes. One main critique of loss aversion suggests that loss aversion does not exist, or is strongly reduced, when the values for losses and gains are symmetric[25,33] (rather than assuming loss aversion and therefore using a larger range for gains than for losses). Our experimental design uses a symmetric range for gains and losses (by adding a constant to all payoffs of a payoff matrix). Although our study was not designed to specifically test this question, we also find no consistent loss aversion in a symmetric design, in agreement with recent developments.

Our experiments focused on social dilemmas. How might our findings apply beyond social decisions? While there is some reason to believe that social and non-social decisions rely (partly) on separate neural mechanisms[3,8,39], empirical evidence from other studies aligns with our results that people try to avoid losses rather than to maximize gains: First, in multi-step decision-making tasks, people's choices were explained by policies to avoid of an outcome of 0 energy points, which was framed as virtually losing one's life[40–42]. Second, in moral decision-making experiments broadly analogous to ours[43], Berman and Kupor separated harm avoidance from harm aversion and found that people preferred avoiding harm. Third, a recent study by Evangelidis[44] found evidence for "rapidly diminishing returns" (for gains and losses). The concept of discrete sensitivity is almost identical to our intercept, although it is theoretically distinct (loss avoidance allows for diminishing returns in addition to a sharp increase around 0). Taken together, it thus seems that people try to avoid negative consequences rather than merely minimizing them, independent of whether it is a moral, ecological, or financial decision. Future studies could more directly translate our approach to these other types of decisions (including the probabilistic gambles paradigm) to test how far our findings apply to other decision-making contexts.

Our results and their alignment with studies from other domains suggest that loss avoidance might be a generic principle underlying many types of decisions. Specifically, an intercept for losses suggests a heuristic such that any loss is devalued by a loss avoidance 'utility fine.' Although our studies focus on social dilemmas, our methods could be easily adapted to test for loss avoidance in other social or non-social contexts (while controlling for loss aversion), thus contributing to the literature on choice, valuation, and preferences, as well as the study of cooperation and social interactions.

## Limitations

The main limitation of this study is that we cannot exclude the possibility that another explanation might account for some of the effects reported here: within Prospect Theory[9,10], many studies include exponents in their modelling, which we simplified here (see Methods/Simulations). For most ranges of exponents typically reported, our simplifications hold, but it is possible that unusually low non-linear components could also explain the results we attribute to loss avoidance. In the normally reported parameter range, however, our results look exactly like loss avoidance. Another limitation is that the iterated experiments were incentivized, but the one-shot experiments were hypothetical, such that it is not entirely clear whether the differences in effect sizes between the two types of social dilemmas are due to the lack of interactive history between both players or due to the difference in incentivization. It seems unlikely, however, that the larger effects in the one-shot games would be reduced or even reversed if the outcomes of such decisions had not been hypothetical but incentivized.

A further question for future research lies in estimating the parameters of our model. The current experimental design was optimised for testing whether there was evidence for (or against) loss avoidance. Future studies could build on our design and optimise the stimuli to be able to estimate the parameters for intercepts, slope, and exponents at the individual level. This would provide a more nuanced perspective into individual variation of loss avoidance (intercept), and its relationship to loss aversion (slope) and diminishing sensitivity (exponent).

## Data availability

All data collected for this study can be found at: https://github.com/dnhi-lab/losses_gains_2x2 and archived on Zenodo: https://zenodo.org/records/15667770.

## Code availability

All code used for the simulations, analyses, and figures mentioned in this manuscript can be found at: https://github.com/dnhi-lab/losses_gains_2x2 and archived on Zenodo: https://zenodo.org/records/15667770.

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

## Acknowledgements

We would like to thank the entire DNHI-lab for comments and discussions, as well as Cristian Ioan, Antonia Wesseloh, Marylin Mintah, and Kira Diermann for assisting with the data collection in Hamburg. The funders had no role in study design, data collection and analysis, decision to publish or preparation of the manuscript. Both authors were supported by the Emmy Noether Research Group grant (392443797) from the German Research Foundation (DFG) to CWK. For the publication fee we acknowledge financial support by Heidelberg University.

## Author contributions

Conceptualization: B.J.K.S. and C.W.K. Methodology: B.J.K.S. and C.W.K. Data collection: B.J.K.S. Analysis: B.J.K.S. Visualization: B.J.K.S. Supervision: C.W.K. Writing—original draft: B.J.K.S. Writing—review & editing: B.J.K.S. and C.W.K.

## Funding

## Competing interests

The authors declare no competing interests.
