## [Transparent Peer Review file · Communications Psychology]

Loss avoidance during social interactions

Corresponding Author: Dr Benjamin Kuper-Smith

Version 0:

Decision Letter:

Dear Dr Kuper-Smith,

Thank you for submitting your manuscript titled "Loss avoidance during cooperation" to Communications Psychology. We have given the paper our careful consideration and find it of potential interest. However, due to certain shortcomings we are concerned that sending the current manuscript out to review could lead to unnecessary delays and quite possibly an undesirable outcome of the review process.

In particular, for manuscripts that interpret null results, we require Bayes Factors or equivalence tests to interpret the null results. For instance, the conclusions regarding the absence of loss aversion in your experiments will have to be supported by the appropriate statistics. Please also ensure to use appropriate language to describe the results. Statements such as 'There is no difference between x and y.' or 'X does not affect Y.' must be revised to read 'We found [no/little] credible evidence of a difference between x and y.' or 'We found [no/little] credible evidence that X affects Y.'.

It is our policy that authors must disclose all deviations from the preregistered protocol and explain the rationale for deviation (e.g., flaw, feasibility, suboptimality). In cases of deviation from the preregistered analysis plan for reasons other than fundamental flaw or feasibility, the originally planned analyses must also be reported. You can find our full policy on preregistration here: <https://www.nature.com/commpsychol/submit/preregistration>

Please add a comprehensive table of all deviations from the preregistration, the reason for the deviation, and where the original planned analysis is reported. To be clear, it is okay to also report the analyses that you think are more appropriate in addition to the preregistered analysis provided the deviation is clearly flagged and the preregistered analysis is included. Only in cases where it is not feasible to run the planned analysis, for example, because it violates the assumptions of the planned statistical test, is it not necessary to report the analysis, but this reasoning must be reported.

We would therefore like to invite you to revise your manuscript to address these concerns before we make a final determination on whether to send your manuscript for external review.

We shall hope to receive your revised version as soon as you are able to complete the suggested revisions. If something similar is published in the interim we will have to consider the impact it has on the novelty of a revised manuscript.

If you anticipate a delay of more than four weeks, please let us know. Should your manuscript be substantially delayed without notifying us in advance and your article is eventually published, the received date may be that of the revised, not the original, version.

We also ask that you ensure your manuscript complies with our editorial policies and reporting requirements.

To that end, we require revised manuscripts to be accompanied by two completed items: a reporting summary that collects information on study design and procedure, and an editorial policy checklist that verifies compliance with all required editorial policies.

- <https://www.nature.com/documents/nr-reporting-summary.zip>>Nature Research Reporting Summary

- <https://www.nature.com/documents/nr-editorial-policy-checklist.pdf>>Editorial Policy Checklist

All points on the policy checklist must be addressed. Your revised manuscript can only be sent to referees if these checklists are completed and uploaded with the revision.

If you are not interested in submitting a suitably revised manuscript in the future please let me know immediately so we can close your file. If you have any questions, please contact me.

Please use the link below when you are prepared to resubmit.

Link Redacted

Thank you for your interest in Communications Psychology.

Best regards,
Troy Lui

Troy Lui, PhD
Associate Editor
Communications Psychology

Version 1:

Decision Letter:

Dear Dr Kuper-Smith,

Thank you for your patience during the peer-review process. Your manuscript titled "Loss avoidance during cooperation" has now been seen by 3 reviewers, and I include their comments at the end of this message. They find your work of interest but raised some important points. We are interested in the possibility of publishing your study in Communications Psychology, but would like to consider your responses to these concerns and assess a revised manuscript before we make a final decision on publication.

We therefore invite you to revise and resubmit your manuscript, along with a point-by-point response to the reviewers. Please highlight all changes in the manuscript text file.

Editorially, we consider it crucial that the results of the revised manuscript are supported by appropriate analysis methods on a par with the state-of-the-art in behavioural economics. To this end, please ensure that the methodological concerns raised by the reviewers, such as the choice of Chi-square test over GLMs and the cooperation rates, are thoroughly addressed. Please also make sure that all the methodological details are reported clearly.

As a reminder, please ensure all deviations from the preregistration are reported. All originally preregistered hypotheses and analyses should be included, unless scientifically unsound, in which case the deviation needs to be highlighted and explained. Please make sure that throughout the manuscript, it is clearly stated which analyses are preregistered and which are exploratory; clarity on this issue is particularly important in instances where the results of preregistered and exploratory analyses deviate. The full policy is here: <https://www.nature.com/commpsychol/editorial-policies/preregistration-policy>

I am attaching an Editorial Requests Table that details critical reporting requirements for the revised manuscript. Please attend to each item and ensure your manuscript is fully compliant. If your revised manuscript is not aligned with these requests on major issues, such as those concerning statistics, it may be returned to you for further revisions without re-review.

Please submit the following items:

- Revised manuscript
- Point-by-point response to the referees' comments
- Cover letter (as a separate document)
- <https://www.nature.com/documents/nr-reporting-summary.zip>>Nature Research Reporting Summary

- [Editorial Policy Checklist](https://www.nature.com/documents/nr-editorial-policy-checklist.pdf)
- Completed Editorial Request Table (attached).

via this link: Link Redacted .

Additional guidance is available in our style and formatting guide [Communications Psychology formatting guide](https://www.nature.com/documents/commspsychol-style-formatting-guide-accept.pdf).

Best regards,

Troby Lui

Troby Lui, PhD
Associate Editor
Communications Psychology

REVIEWER EXPERTISE:

Reviewer #1: loss aversion

Reviewer #2: cooperation

Reviewer #3: loss aversion, cooperation

REVIEWER REPORTS:

Reviewer #1 (Remarks to the Author):

In seven experiments the authors try to disentangle the effect of loss avoidance and loss aversion in social dilemmas. The results show that participants exhibit a tendency towards loss avoidance but not loss aversion. I think these results are highly interesting and that overall the paper is well thought about and organized. Still, in places the clarity of the analysis can be improved and some important details are missing. I should note that I was a reviewer of the paper for a different journal.

Major comments:

1. The authors' exclusion criteria are unclear and my view are not sufficiently justified as currently stated ("Participants were excluded if they gave too many late/invalid responses or had taken part in one of the previous experiments"). The number of those excluded are missing and the exact criteria are not clear in the present manuscript. In my view, no participants should be excluded based on what seems like post-hoc considerations. Especially, as far as I could see, the items for which participants were excluded were not described in the pre-registration protocol, nor is the criterion for success/failure mentioned. This is especially pertinent given the apparently large number of excluded participants (as indicated in the prior version of the manuscript). Why not include all participants?

2. (a). The assumption in the simulation of a 50:50 chance that the other player will cooperate or defect seems simplistic. Why should this be the case? Please include a robustness analysis in the model clarifying its predictions for different circumstances. Does the predicted difference remain under different assumptions? (perhaps you can add this to the simulation and summarize in a sentence or two in the main text). (b). Some more detail should be added regarding the specific loss aversion and loss avoidance models used. I assume there is a parameter governing the degree of loss aversion and loss avoidance. What was it set to? You only describe the exponential parameter in the main text. Please describe all model parameters and their values. (c). Generally speaking I think the simulation should be fully described the supplementary section.

Minor comments

Abstract: I think the abstract should be somewhat more cautious regarding your findings on loss avoidance. For the stag-hunt game you find evidence for loss avoidance in one study but not in the other.

You cite Brown et al. (2024) as supporting loss aversion. I suggest you also cite a meta-analysis based on the same studies which shows that the loss aversion in Brown et al. (2024) is driven by studies using smaller losses than gains and an order effect, with no loss aversion being found for estimates based with symmetric gains and losses (Yechiam, E., & Zeif, D. (2025). Loss aversion is not robust: A re-meta-analysis. *Journal of Economic Psychology*, 107, 102801).

Lines 52-53: The way you define loss aversion and loss avoidance in the beginning of this paragraph is not clear. Loss aversion is not the tendency to minimize losses. This is simply a feature of expected value and expected utility. The same goes for loss avoidance.

Page 9: Please translate the name of the first ethics committee (it comes out weird that one committee name is translated and the other is not).

Figure 3: How did you aggregate (combine) averages across studies? A simple mean (of means)?

Reviewer #2 (Remarks to the Author):

I have read the manuscript "loss avoidance during cooperation" with great interest. This research seeks uses a number of social dilemma games to identify (a) loss avoidance and (b) loss aversion in cooperation. The experiments are thorough and provide robust evidence on the key questions. The open science practices, in particular, are exemplary. I only have one question regarding the modelling approach, but otherwise believe this manuscript is ready for publication in its current form.

Scope of this review

The authors provide access to data, code, and materials. Since the analyses were conducted in Matlab, I did not check for computational reproducibility; I did successfully replicate key results from Study 5 in R. The Matlab analysis files contain labels for the variables in the data files -- it would be helpful to note that in the readme file. I inspected the preregistrations and observed no undeclared deviations.

Major comments

My main question regards the best way to model the data from experiments 4-7. The manuscript reports two kinds of tests: separate Chi-square tests and GLMs. For the GLMs, there are three models: without expectations, with main effect for expectations, and with interactions with expectation. The manuscript prioritises the results from the Chi-square tests. Depending on the specification of the GLMs, they are inconsistent with the Chi-square tests. For example, for Experiment 5, the Chi-square tests support loss avoidance, but not loss aversion in the PD. This is consistent with the GLM results when controlling for expectations, but in the GLM without expectations. It is not clear to me why Chi-square tests are prioritised in the manuscript; at least for experiments 6 and 7, both are preregistered.

More importantly, the manuscript reports GLMs controlling for expectations. However, unless I misread the manuscript, expectations are post-treatment variables. Controlling for these is problematic because it violates random assignment. I would therefore suggest focusing on GLMs without controls for expectations (if one wanted to control for expectations, some form of strategy method approach would have been preferable).

Minor comments

p. 4: I found the simulations difficult to interpret without knowing the aversion/avoidance parameters. This is explained in the methods, but perhaps it would be possible to add a sentence on p. 4 giving at least some intuition?

p. 6: One terminological quibble: you sometimes write "loss avoidance significantly reduced cooperation" (p. 6), or similar phrases. I don't think that's quite accurate -- loss avoidance is what's inferred from the lower cooperation rate. Perhaps a formulation such as "evidence for loss avoidance, indicated by lower cooperation..." would be more accurate.

p. 9: "Stag-Hunt" should be written without a hyphen

p. 9: "always reduce cooperation" -> "always reduce cooperation in the PD"

I always sign my reviews.

Simon Columbus | simon@simoncolumbus.com

Reviewer #3 (Remarks to the Author):

I like the design and evidence in this paper quite a lot. However, the analysis and hence all the tentative conclusions are either clearly not quite correct or not SOTA.

The basic research question is whether affine transformations of payoffs in a 2-person 2-strategy game theory matrix (i.e. multiplying by $a > 0$ and adding b) make a difference to behavior. Specifically, the authors note (building on refs 21-24) that if

certain transformed payoffs cross the boundary from positive to negative that could induce changes because of avoidance of losses (a “utility fine”—a great phrase-- $u(x)=x-b$ for $x<0$) or loss-aversion. This an idea that has been shown in a few experimental economics papers (those carefully cited 21-24) but not picked up in psychology at all. So this is a welcome importation of that idea to a psychology audience, with a much more adventurous multi-game design, and by two talented social neuroscientists who may be able to even more with this idea including in circuitry identification.

(1) **However, the analysis of the results for some experiments is made complicated by the fact that in experiments 1-3 subjects play repeated PDs multiple times with random (with replacement?) partners in 6-person groups, for 12 payoff matrices. Obviously there are many dynamic influences so these choices, for one person, cannot be treated as iid observations. Test statistics always assume some degree of independence. You need to carefully figure out what statistics to use. For example, players may learn or change over time. The players are also interdependent (e.g. one six person group in which several people always choose D will lead many of them to choose D). So you can't treat each person in a six-person group as having six different independent cooperation rates. I do not know how to best control for these sources. It is likely that the inflated results from the repeated PD experiments (which do not replicate well in one-shot experiments) are associated with these statistical issues.

a. Frechette (2011) is an old paper on “session effects” (which applies to your 6-person groups as a “session”) and may be helpful as a start.https://papers.ssrn.com/sol3/papers.cfm?abstract_id=1937814

(2) A major concern is simple to resolve: How do you get cooperation rates near 50% in the PD for all-positive or all-negative payoffs? The expected value of D is always higher than C for all probabilities p (expected cooperation); so how do you get high predicted cooperation rates? Are you averaging cooperation of one player (e.g. row) computed from expected utilities (lines 117-118) with the assumed 50% of column play? That does not make sense because presumably the column player calculation will be the same as row.

a. More generally, it doesn't seem like you have done the proper game-theoretic analysis. In the SOTA analyses of these games, you treat row and column players (since both are human) as making similar calculations (though of course the payoffs may be different) and assuming some p% chance of the other player's behavior. But in the usual benchmark analysis, the value of p is derived endogeneously—i.e. the row player computes utility based on p(column) and column chooses based on p(row). You can then either let p(row) as guessed by the column player be 50% (or some fixed number) but it obviously ignores the fact that the subjects in the experiment may be computing a guess about p from what they think row players are computing.

(3) The SOTA way to analyze these data is like so, for the one-shot players with no feedback: Start with these lines in the paper

Lines 117-118 contain a nice simple formula for understanding avoidance and aversion (using $U(x)$ to reflect utilities instead of y)

$$U(x|x>0)=x$$

$$U(x|x<=0)=ax-b$$

Now loss-avoidance alone is $b>0$.

Loss-aversion is $b=0, a>1$

You can test a hybrid where $b>0$ *and* $a>1$

In one version you assume b and a are the same for everyone and just pile in all the one-shot game data together. Utilities are created above and weighted by a perceived P(others) about what others do. You can set that equal to what others actually do (imposing an equilibrium assumption) or use expectation, or some other idea (Note that because they're playing once with no repetition the statistical concerns at ** above do not apply at all.) You can then estimate the values of a and b across the population. That would be very interesting. This is called MLE—compute a “likelihood function” P(cooperate) from all those moving parts, and find parameter values (a,b) that maximize how well the prediction fits the data.

(4) You also seem to measure “expectations” (e.g. Table S2 refers to it). But methods and Supplemental say nothing about it.

(5) I can't find the Methods/Simulations that are referred to in the reviewer zip file. How exactly were the simulations done?

(6) What the supplemental Tables report is mysterious. Take Table S2. What are M1 and M2? What are the percentages in the Table? What test produces the t-statistics? Please be more clear on details

(7) Materials and methods does not mention anything about stake size

(8) Line 478. Can you do something with gender and age data? E.g. use them as covariates for predicting behavior. If they do have effects on choice then including them will increase the power to estimate the true effects of loss on behavior.

(9) Table S1 what are the a and b values? Table note does not explain. Is it related to the a and b the in line 118 model?

(10) The title is not quite right because not all these games involve cooperation (e.g. chicken is mixed-motive).

(11) Lines 159+ these $d=.4$ effects are very large. I suspect you are not estimating them correctly (see comment * above about standard errors)

(12) Lots of nice features of this paper to praise!

a. Line 456 careful Prereg analysis is terrific. That is so helpful for reviewers and it is exactly how Prereg should work, if authors make the extra effort to clarify any deviations. Thank you for that. Figs 1-2 are really good. A lot of information is conveyed there.

* TRANSPARENT PEER REVIEW: Communications Psychology uses a transparent peer review system. This means that we publish the editorial decision letters including Reviewers' comments to the authors and the author rebuttal letters online as a supplementary peer review file. However, on author request, confidential information and data can be removed from the published reviewer reports and rebuttal letters prior to publication. If your manuscript has been previously reviewed at

another journal, those Reviewers' comments would not form part of the published peer review file.

Version 2:

Decision Letter:

Dear Dr Kuper-Smith,

Your manuscript titled "Loss avoidance during social interactions" has now been seen by our reviewers, whose comments appear below. As Reviewer #3 had become unavailable, we asked Reviewer #2 to comment on the issues raised by Reviewer #3 (also appended below). In light of their advice I am delighted to say that we are happy, in principle, to publish a suitably revised version in Communications Psychology.

We therefore invite you to revise your paper one last time to address the remaining concerns of our reviewers and a list of editorial requests. At the same time we ask that you edit your manuscript to comply with our format requirements and to maximise the accessibility and therefore the impact of your work.

EDITORIAL REQUESTS:

SUBMISSION INFORMATION:

OPEN ACCESS:

* **CODE AVAILABILITY:** All Communications Psychology manuscripts must include a section titled "Code Availability" at the end of the methods section. We require that the custom analysis code supporting your conclusions is made available in a

publicly accessible repository at this stage; please choose a repository that generates a digital object identifier (DOI) for the code; the link to the repository and the DOI must be included in the Code Availability statement. Publication as Supplementary Information will not suffice.

*** DATA AVAILABILITY:**

Link Redacted

Best regards,

Troy Lui

Troy Lui, PhD
Associate Editor
Communications Psychology

REVIEWERS' COMMENTS:

Reviewer #1 (Remarks to the Author):

I think the paper has markedly improved. The exclusion criteria and numbers are transparently presented and my other comments have been addressed.

Reviewer #2 (Remarks to the Author):

Thank you for the opportunity to review this revised manuscript. The authors have fully addressed my earlier comments; I have no further objections and believe the manuscript is suitable for publication.

I always sign my reviews.
Simon Columbus | simon@simoncolumbus.com

I was asked to comment on the authors' response to Reviewer 3. I will focus here on questions R3.3(a) and R3.4, which seem to me the most fundamental criticisms. In both cases, the authors argue that the proposed approach goes beyond the scope of the paper.

First, comment R3.3(a) suggests an alternative approach to deriving predictions, based not on simulations but on a game-theoretic analysis of belief formation. I agree that this would potentially be interesting, but it strikes me as extending quite a bit beyond the current scope of the paper. Importantly, this would require a commitment to some account of belief formation which wouldn't be testable on the extant data. I think it is fair to stick to the more descriptive simulation approach.

Second, comment R3.4 is, as the authors point out, somewhat cryptic. In my reading, it suggests two things, namely (a) jointly modelling loss aversion and loss avoidance and (b) estimating values for each parameter. I had a similar reaction when I initially read the manuscript -- this is what I would have expected given the question this paper asks. As the authors rightfully argue, this isn't sensible given the data. In this case, I think the manuscript makes a sufficient contribution anyway, even if additional data and modelling could provide further interesting insights. I suspect that Reviewer 3 may not agree with me here, but this seems to me a question more about taste than about accuracy.

I always sign my reviews.
Simon Columbus | simon@simoncolumbus.com

Reviewer comments in blue

Author responses in black

Text from manuscript indented, in different font, with additions and ~~deletions~~.

For easier communication and referencing other comments, we re-numbered all reviewer comments into one coherent scheme (**Rx.y**, where x is the N of the reviewer, and y is the N of comment for that reviewer). Apart from this, all reviewer comments are shown exactly as in the original peer review file.

We would like to thank all reviewers for taking the time to provide such insightful comments. The new version of our manuscript tries to address all reviewer comments, and it has certainly improved as a direct consequence of the work you have put into reviewing it.

REVIEWER REPORTS:

Reviewer #1 (Remarks to the Author):

R1.1 In seven experiments the authors try to disentangle the effect of loss avoidance and loss aversion in social dilemmas. The results show that participants exhibit a tendency towards loss avoidance but not loss aversion. I think these results are highly interesting and that overall the paper is well thought about and organized. Still, in places the clarity of the analysis can be improved and some important details are missing. I should note that I was a reviewer of the paper for a different journal.

Thank you very much for reviewing our manuscript (again) and for your considerate comments. We hope that we address all your points in the following.

Major comments:

R1.2 The authors' exclusion criteria are unclear and my view are not sufficiently justified as currently stated ("Participants were excluded if they gave too many late/invalid responses or had taken part in one of the previous experiments"). The number of those excluded are missing and the exact criteria are not clear in the present manuscript. In my view, no participants should be excluded based on what seems like post-hoc considerations. Especially, as far as I could see, the items for which participants were excluded were not described in the pre-registration protocol, nor is the criterion for success/failure mentioned. This is especially pertinent given the apparently large number of excluded participants (as indicated in the prior version of the manuscript). Why not include all participants?

We completely agree that excluded participants should be made clear. To respond to this comment, we'll first provide general clarifications with associated changes to the text, and then break down the details experiment-by-experiment.

We provide links to the preregistrations, which we follow almost completely to the letter, and we include a section called 'Deviations from preregistrations' (not to mention we share the code publicly that excludes participants); this information is available on pages 16-17.

We clearly state how many participants we tested and how many we included in the final sample (Materials & Methods / Demographics). While we did not explicitly mention the number of excluded participants, this can be calculated instantly from Table 3 in the Demographics section (lines 560-562):

Experiment	N included (N total)	Age mean (SD)	Gender % male/female/diverse/prefer not to say
1	24 (24)	28 (8)	42/58/NA/NA
2	46 (48)	27 (5)	39/61/NA/NA
3	44 (48)	27 (7)	41/59/NA/NA
4	83 (93)	25 (5)	44/56/NA/NA
5	447 (902)	39 (12)	51/47/2/1
6	603 (751)	41 (13)	51/47/1/1

7	603 (754)	40 (13)	50/49/1/0
---	-----------	---------	-----------

Table 3 - Demographics for all experiments. Experiments 1-4 include 'NA' in the gender column because we only included 'female' and 'male' as response options.

Crucially, we're not excluding participants for arbitrary and constantly shifting reasons, it's always the same reasons: we exclude participants who 1) we think didn't understand the task, and 2) who took part in previous experiments. The exclusion criteria were always of the same nature for each type of experiment: in the iterated experiments, we excluded participants if they didn't respond adequately (i.e., too slow or with the wrong button keys) and in the one-shot experiments, we excluded anyone who made errors in the test questions. Both of these exclusions are purely about ensuring that the participants that we do include actually understand the task and are able to respond adequately. While in the laboratory we had to exclude almost no one due to lack of understanding (due to the length of the experiment, it made sense to explain any misunderstanding to participants), in the one-shot experiments, it was financially more sensible to ignore anyone who didn't understand the task, rather than pay everyone twice the amount due to an increased average study duration. In the iterated experiments, we also excluded participants due multiple participation, but this again seems like a completely uncontroversial exclusion criteria, given that the experiments were supposed to be replications with independent samples. In any case, these exclusion criteria are stated explicitly in the manuscript and the preregistrations.

But we agree that a lot of this information was lacking in the original version of the manuscript. To clarify these points (including the threshold for exclusions), we added the following clarification to the Demographics section, which now starts with (lines 521-530):

For all experiments, we excluded participants (see Table 3) for two reasons: 1) if we thought that participants might not have understood the task properly, and 2) due to taking part in multiple experiments. These exclusions were always done in the same manner: in the iterated experiments, we defined a participant as not understanding the task properly if they failed to press the appropriate keys in the required time more than 10% of all trials, and in the one-shot experiments, it was defined by making even a single error in the test questions. We excluded such participants. Additionally, in the iterated experiments, due to administrative error, 3 participants (1 in Experiment 2, 2 in Experiment 3) were excluded because they had taken part in one of the previous experiments. We excluded these participants because Experiments 2 and 3 were intended to be replications and extensions with independent data.

We also made the following changes to further highlight how and why we excluded participants (lines 538-559):

The participants for Experiments 5-7 were recruited via Prolific and tested on SoSciSurvey.de, a web platform for collecting online data. In Prolific, we only included participants from the UK who were native English speakers, with a prior approval rate of >95% (to pre-emptively exclude unserious participants); we pre-emptively excluded anyone who participated in any of our previous experiments, to remove the possibility of participants appearing in multiple datasets. We used Prolific's 'balanced gender' feature, such that there were equal number of male and female participants in each sample. Slight variations in each experiment and condition can exist due to random allocations into groups, and due to failing comprehension tests.

For Experiments 1-3, participants received detailed instructions with test questions, and experimenters answered any questions. Participants were excluded (see Table 3 for N of participants total and included in all experiments) if they gave too many (>10%) late/invalid

responses or had taken part in one of the previous experiments (due to administrative error, a few one participants in Experiments 2 and two participants in Experiment 3 had taken part in Experiments 1 or 2, and we excluded their later participation).

For Experiments 4-7, which were very short (< 5 minutes), we provided an explanation of the task and a few test questions. We excluded participants if they made any errors in the test questions (three test questions in Experiment 4, four test questions in Experiment 5 and ~~three~~ two test questions in Experiments 6 and 7). Alternatively, we could have offered further explanation and test questions to participants who made an error in the test questions in the one-shot experiments, but this would have increased the average study duration. Instead, we opted for the financially more efficient route of excluding anyone who might not have understood the task.

For the reviewer, we also clarify the exclusions for each experiment in further detail. We try to address this comment succinctly, but this is somewhat difficult, given that the comment doesn't specify which experiment the criticism applies to. We'll go through it experiment by experiment.

For **Experiment 1**, we don't exclude anyone.

For **Experiment 2**, we exclude 2 participants. The preregistered exclusion criteria are:

Data exclusion

No data exclusions will be made other than:

- Exclude trials on which:
 - Participants made an invalid response (up/down instead of left/right or vice versa)
 - Participants didn't respond fast enough (i.e., they didn't press a button within the 6s time window)
- Exclude a participant's data from the analysis if: more than 10% of invalid or late responses

Figure 1 - The preregistered exclusion criteria for Experiment 2. Screenshot from <https://osf.io/dzkqg>.

We stuck precisely to this, with one exception. As specified in the section 'Deviations from preregistrations' we also excluded 1 participant because they had accidentally been allowed to participate in Experiment 1 and in Experiment 2 (this was due to changing student assistants whom we hadn't adequately instructed to check new participants for prior participation). Given that the purpose of this experiment was to provide a replication of Experiment 1 in a new data set, it made no sense to us to adhere blindly to an underspecification in our preregistration.

For Experiment 3, we excluded 4 participants. This was done by precisely following the following preregistered exclusion criteria:

Data exclusion

No data exclusions will be made other than:

- Exclude trials on which:
 - o Participants made an invalid response (up/down instead of left/right or vice versa)
 - o Participants didn't respond fast enough (i.e., they didn't press a button within the 6s time window)
- Exclude a participant's data from the analysis if:
 - o More than 10% of invalid or late responses
 - o They claim to not have understood the experiment (evaluated in a post-experiment questionnaire)
 - o They correctly identify the hypothesis (evaluated in a post-experiment questionnaire)
 - o They knew who they played against in each round (evaluated in a post-experiment questionnaire)
 - o They already took part in a previous experiment from this study

Figure 2 - The preregistered exclusion criteria for Experiment 3. Screenshot from <https://osf.io/z4geq>.

For the one-shot experiments, we excluded anyone who made an error in the test questions (see below for details for each experiment). This was done to ensure that the participants in our final sample actually understood the task. We made no other exclusions.

For Experiment 4, we excluded 10 participants. This experiment was administered by various student assistants, who simply handed a sheet of paper to their participants at the end of other (non-social) experiments at the institute. This sheet contained three test questions, and we excluded any participant who made even a single error on the test questions. Because this experiment featured a single decision of the participants that made the hypothesis relatively easy to guess if one understood the categories, we were concerned that experimenter bias might influence participants' behaviour. We therefore didn't explain the experiment to the student assistants, who were instructed to hand the participants the sheet with no further explanation.

For Experiment 5, we excluded 455 participants, about half of the sample. This is, to us, the only potentially contentious exclusion of participants. But again, it follows the same logic as all other one-shot experiments: we provide the instructions and some test questions, and we then exclude anyone who made even a single error. The high number of exclusions in this experiment is likely due to our instructions being confusing (which is why we improved them for Experiment 6).

Experiment 5 was not preregistered and we exclude a large proportion of participants, but this was due to practical reasons: because the experiment involved only a single experimental question, it was cheaper to just let everyone answer the experimental questions and exclude anyone who made a mistake, rather than explaining the instructions again and asking further test questions (which would increase the average completion time on Prolific, which would force us to pay everyone more). Thus, we decided a priori to exclude anyone who made a single error on the test questions (as with Experiments 4, 6, and 7). We never analysed the data any other way. We were surprised by how many people we had to exclude, and we improved the instructions for Experiments 6 and 7 (and reduced the test questions from four to two).

We also don't see such a big problem in excluding more participants than necessary in Experiments 4 and 5 because those are (by not being preregistered) somewhat exploratory in nature; to ensure readers understand this, we call them the 'laboratory pilot' (Experiment 4) and the 'online pilot' (Experiment 5) in lines 241-242. The results from Experiment 5 replicate perfectly in the Prisoner's Dilemma data for the preregistered Experiments 6 and 7, such that it should be clear that our results do not depend on the precise exclusion of participants in one particular study.

For Experiment 6, we excluded 148 participants, following the preregistered exclusion criteria precisely:

Outliers and Exclusions

In our experiment, we include 2 test questions. We will exclude anyone who doesn't answer both test questions correctly. Participants will receive the tests questions with the payoffs of the decision of the condition/category they're in

Figure 3 - The preregistered exclusion criteria for Experiment 6. Screenshot from <https://osf.io/72fj6>.

For Experiment 7, we excluded 151 participants, following the preregistered exclusion criteria precisely:

Outliers and Exclusions

In our experiment, we include 2 test questions. We will exclude anyone who doesn't answer both test questions correctly. Participants will receive the tests questions with the payoffs of the decision of the condition/category they're in.

Figure 4 - The preregistered exclusion criteria for Experiment 7. Screenshot from <https://osf.io/p6xdr>.

We could reanalyse the data from the one-shot experiments without excluding so many participants, but it's not clear what such a re-analysis would achieve: if our results remain, we were unnecessarily cautious; if the results no longer hold after including participants who we think may not have understood the task, then this could be used as evidence that the participants did indeed not understand the task properly (which would lead to more noise in the data).

Again, we understand that the different number of excluded participants in our experiments might be a cause for concern, but we hope to have shown (and clarified in the manuscript) that this was not done in a capricious and post-hoc manner but always using the same criteria (even if the specific implementation might differ slightly between experiments).

R1.3(a). The assumption in the simulation of a 50:50 chance that the other player will cooperate or defect seems simplistic. Why should this be the case? Please include a robustness analysis in the model clarifying its predictions for different circumstances. Does the predicted difference remain under different assumptions? (perhaps you can add this to the simulation and summarize in a sentence or two in the main text).

The initial simplified assumption that the other player would cooperate with a 50% chance was based on the empirical observation that (roughly) 50% of people cooperate in the Prisoner's Dilemma. We therefore took this as the starting point for our simulations. As specified in lines

602-604, we confirmed that our hypotheses still hold when using not a blanket 50% expectation across all games and categories, but the empirical expectation of what our participants thought the other player would do (estimated in Experiment 7), separately for each game and each category (we only measured this in the one-shot games).

Having said that, there are some boundary cases that are almost tautological: our hypotheses are about asymmetries between possible losses and gains. Thus, at the extreme ends of p , where losses are either guaranteed or impossible, our hypotheses break down for some conditions. We added this clarification to the simulation section (lines 602-611):

Second, we assumed that players expect a 50:50 chance that the other player will cooperate or defect (as a control, we later inserted participants' estimated probability of what the other person would do, which led to indistinguishable predictions). Our simulated predictions require mixed outcomes and therefore do not always hold at the extreme ends of p (i.e., near 0 and 1): if a player knows that the other will cooperate or defect, many gain/loss asymmetries no longer apply. For example, if a player in the Prisoner's Dilemma Category 2 knows with certainty that the other person will cooperate, then the only loss outcome of the payoff matrix (player one cooperates and player two defects) is not possible for player one, which makes any attempt to avoid a loss unnecessary. In other words, our hypotheses require the possibility of losses and gains for asymmetries between the two to become relevant for people's decisions (see Figure S3).

We also added the following figure to the supplementary materials to show this graphically:

Figure S3 – How different probabilities of what the other person will do affect the predicted cooperation rates. These examples use the Prisoner's Dilemma, but the same principles apply to Stag Hunt and Chicken, too. A) The predicted cooperation rates for loss avoidance and loss aversion, assuming a 50% chance the other player will cooperate. B), C), and D) show how the cooperation rates change as the expectation that the other will cooperate go from 50% to 100%. This shift leads to a shift in where the effect happens (e.g., for loss avoidance, with 50% expectation, the effect is of equal magnitude for C2 and C4; but with certainty that the other will cooperate, the effect exists only for C4, but much larger). E) shows the predictions with certainty that the other will defect, with a reverse pattern observed to the pattern show in D. These shifts show that under extreme cases certain hypotheses can break down: our

hypotheses are about potential gain-loss asymmetries, so some hypotheses fall apart if losses or gains become increasingly (un)likely. For example, in C2, if a player knows the other will cooperate, the only negative outcome is no longer possible (because the other player will not defect). F) In Experiment 7, we asked participants to rate the probability that the other person would cooperate. The mean responses were close to 50% (range: 40%-58%). If we use the mean expected cooperation rate of the other player (separately for each category), then the predictions are practically indistinguishable from the 50% simplification.

We hope that these additions clarify that the seemingly simplistic assumption that the other person will cooperate with a 50% chance simplifies our simulation without a loss of generality (apart from extreme cases that turn mixed situations into pure gain or pure loss situations).

R1.3(b). Some more detail should be added regarding the specific loss aversion and loss avoidance models used. I assume there is a parameter governing the degree of loss aversion and loss avoidance. What was it set to? You only describe the exponential parameter in the main text. Please describe all model parameters and their values.

Yes, this comment was made by all reviewers (see also **R2.5** and **R3.6**). Because our simulations lead to categorical predictions, the precise values that we chose for loss aversion and loss avoidance merely determine the magnitude of the effect, rather than the pattern (again, extreme values can lead to the simulation to break down for trivial reasons). We've added this clarification to the beginning of the simulation section in the Methods section (lines 578-589):

The purpose of these simulations was to show the categorical pattern of behaviour that loss aversion and loss avoidance would lead to in our experimental design. For loss aversion, we chose a (or lambda) = 2, which was commonly thought to be a normal value in the population (but that recent meta-analyses suggest are overestimated^{33,34}. Using lower values for loss aversion would lead to the same pattern, but with a smaller effect size). For loss avoidance, we chose b = 4. The magnitude of the parameter mainly influences the magnitude of the effect (e.g., a larger loss avoidance parameter would lead to a stronger reduction of cooperation for Categories 2 and 4 in the Prisoner's Dilemma; a larger loss aversion parameter would lead to a steeper downward shift from Category 1 to Category 5 in the Prisoner's Dilemma). For extreme loss aversion, the patterns break down (e.g., if loss aversion were extremely large, cooperation would be 0% for any payoff matrix with a loss), but within normal ranges, the size of the parameters merely manipulates the magnitude of the effect.

R1.3(c). Generally speaking I think the simulation should be fully described the supplementary section.

While we agree that our descriptions of the simulations were lacking in the initial version of the manuscript (as evidence by multiple question from all reviewers), we're not sure that there is any need to explain it further in the supplementary materials. We believe that we've now fully described the overall procedure of the simulations in the main text, while adding clarifications about our assumptions in the Materials & Methods section. The code for the simulations is publicly available and linked (https://github.com/dnhi-lab/losses_gains_2x2/tree/main/simulations), and even though some people might not be familiar with Matlab, current LLMs like ChatGPT can easily translate it into any other common coding language, if necessary. We're not sure what more we could say about it, without repeating ourselves. If the additions we made with this round of revisions remain insufficient to understand the simulations, we are happy to add specific details, but we feel like we've explained the script verbally in full detail now in the manuscript, leaving nothing more to explain in the supplementary materials.

Minor comments

R1.4 Abstract: I think the abstract should be somewhat more cautious regarding your findings on loss avoidance. For the stag-hunt game you find evidence for loss avoidance in one study but not in the other.

We agree that we could be a bit more cautious here and we deleted the part where we say it applies to all dilemmas. We still believe though that we do find evidence for loss avoidance in Stag-Hunt (all tests in Experiment 6; in Experiment 7 one Chi-squared test was significant, the other two effects were in the same direction but smaller than expected, and the logistic models showed an overall effect when considering all categories at once). We agree that in light of the slightly unclear results from Experiment 7 (which we attribute to a mixture of ceiling effects and requiring most participants to switch cooperation and defection for Categories 2 and 4 after playing Chicken first, which has strong effects and the opposite predictions from Stag Hunt), it is not necessary to make this statement in the abstract. The abstract now reads:

Social interactions lead to outcomes for oneself and others, which can be gains or losses. Yet, it is unclear how exactly people's social decisions are affected by whether an outcome is above or below zero. We systematically varied whether the outcomes of social dilemmas (Prisoner's Dilemma, Stag Hunt, Chicken) were gains, losses, or combinations thereof. Across seven experiments (4 preregistered; $N_{\text{Offline}} = 197$, $N_{\text{Online}} = 1,653$), participants ~~consistently~~ tried to avoid losses altogether (loss avoidance) ~~in all types of dilemmas~~, but there was no consistent evidence that they tried to minimize losses (loss aversion). If cooperation avoided losses, people cooperated more; if defection avoided losses, people defected more, even if this imposed a loss on the other person. Our results suggest that cooperation and social interactions can be influenced systematically if the situation allows people to avoid losses.

R1.5 You cite Brown et al. (2024) as supporting loss aversion. I suggest you also cite a meta-analysis based on the same studies which shows that the loss aversion in Brown et al. (2024) is driven by studies using smaller losses than gains and an order effect, with no loss aversion being found for estimates based with symmetric gains and losses (Yechiam, E., & Zeif, D. (2025). Loss aversion is not robust: A re-meta-analysis. *Journal of Economic Psychology*, 107, 102801).

Yes, thank you; that review came out after we submitted the manuscript and we now added it to the new version; we also added a similar meta-analysis by Walasek & Stewart (2024, *J Econ Psych*) that we became aware of after submitting our initial manuscript to *Communications Psychology*.

Narratively, we present the 'standard' view of loss aversion in the introduction without the recent developments (because our study was not specifically designed to test whether loss aversion exists; it's more of an application to a different context). In the discussion, we highlight that our inconsistent results for loss aversion are not unprecedented, given recent findings. More specifically, your comment made us realise that there is a parallel between one of the main criticisms of how loss aversion is measured (asymmetric ranges for gains and losses artificially inflate loss aversion), and that our inconsistent results for loss aversion also occur in a design with symmetric ranges for gains and losses. In addition to including the citations of the meta-analyses, we also added the following point to the discussion (lines 381-389):

Our mixed results from loss aversion match the mixed results about loss aversion from non-social contexts²⁹⁻³⁴. The field might face a similar situation to the risk elicitation puzzle³⁵, where different ways of measuring the supposedly same phenomenon lead to different outcomes. One main critique of loss aversion suggests that loss aversion does not exist, or is strongly reduced,

when the values for losses and gains are symmetric^{29,33} (rather than assuming loss aversion and therefore using a larger range for gains than for losses). Our experimental design uses a symmetric range for gains and losses (by adding a constant to all payoffs of a payoff matrix). Although our study was not designed to specifically test this question, we also find no consistent loss aversion in a symmetric design, in agreement with recent developments.

R1.6 Lines 52-53: The way you define loss aversion and loss avoidance in the beginning of this paragraph is not clear. Loss aversion is not the tendency to minimize losses. This is simply a feature of expected value and expected utility. The same goes for loss avoidance.

Thank you for this comment. We tried to provide a very brief intuition of the difference between loss aversion and loss avoidance, but your comment made us realise that this has resulted in undue imprecision. To alleviate this problem, we've changed the second paragraph:

But do people try to minimize losses (loss aversion), or might they try to avoid losses entirely, irrespective of magnitude (loss avoidance)? Loss aversion has received most of the attention in the discussion of asymmetries between losses and gains, but another asymmetry, loss avoidance, has received much less attention. Loss aversion is typically defined as a steeper slope for losses than for gains, while loss avoidance^{12,13} is typically defined by an intercept for losses. Loss aversion and loss avoidance have often been conflated, so it is unclear how exactly losses and gains affect people's decisions. In this article, we contrast loss aversion with loss avoidance in social decisions and find evidence for loss avoidance but not for loss aversion.

R1.7 Page 9: Please translate the name of the first ethics committee (it comes out weird that one committee name is translated and the other is not).

We made the change as suggested. The section now reads:

We obtained informed consent to participate from all participants. The lab-based studies were approved by the ethics committee of the medical association (Ärzttekammer) Hamburg (PV5746), and the online studies were approved by the ethics committee of the medical faculty at the University of Heidelberg (S-127/2020).

R1.8 Figure 3: How did you aggregate (combine) averages across studies? A simple mean (of means)?

The aggregation for the figure was done slightly differently for the two types of experiment: for the iterated Experiments 1-3, it is a mean of means: we calculated the mean cooperation rate for each participants in each category, combined them, and calculated the mean (+SEM) of that group, while for the one-shot Experiments 4-7, where there is only 1 decision per game per participant, we combined all those binary choices in each category and each game, and calculated the new mean (+SEM) for each category/game type combination. To clarify this in the text, we added the following description to the figure legend of Figure 3:

Figure 3 - The simulated predictions for each game (left column) and a summary of the (empirical) cooperation rates across all experiments for the five categories (right column). Based on simulations, we predicted cooperation rates for each game for each category, assuming an agent has a utility curve with loss avoidance (top) or loss aversion (bottom). For details of the simulations, see the first section in Results, Figure 2 and Methods. Results: Blue top: iterated Prisoner's Dilemma; blue bottom: one-shot Prisoner's Dilemma; orange: one-shot Stag Hunt; red: one-shot Chicken. Experiments 1-5 used the Prisoner's Dilemma,

Experiment 6 used Stag Hunt and the Prisoner's Dilemma, and Experiment 7 used Chicken, Stag Hunt, and the Prisoner's Dilemma. Experiments 1-3 were iterated, and Experiments 4-7 were one-shot. Experiments 1-4 were offline in the lab, Experiments 5-7 were online. For illustrative purposes, we combined the data from similar experiments here (for the iterated Experiments 1-3, we took the mean cooperation rate of each participant in each category, combined them, and calculated a mean of means for each Category; for the one-shot Experiments 4-7, we combined all (binary) choices for each category for each game, and calculated a new mean response), even though almost all statistical tests were done separately for each experiment. For cooperation rates of individual studies, see SOM. Error bars indicate the standard error of the mean, and the numbers (in parentheses) on the x-axis indicate the number of participants included in each bar.

Thanks again for taking the time to reviewing our article. Your comments have improved the manuscript and we hope we've addressed your comments adequately.

Reviewer #2 (Remarks to the Author):

R2.1 I have read the manuscript "loss avoidance during cooperation" with great interest. This research seeks uses a number of social dilemma games to identify (a) loss avoidance and (b) loss aversion in cooperation. The experiments are thorough and provide robust evidence on the key questions. The open science practices, in particular, are exemplary. I only have one question regarding the modelling approach, but otherwise believe this manuscript is ready for publication in its current form.

Thank you very much for reviewing our article and for your considerate comments. We're happy that you like our article and recommended it to be published. We hope that our following responses adequately address your comments.

Scope of this review

R2.2 The authors provide access to data, code, and materials. Since the analyses were conducted in Matlab, I did not check for computational reproducibility; I did successfully replicate key results from Study 5 in R. The Matlab analysis files contain labels for the variables in the data files -- it would be helpful to note that in the readme file. I inspected the preregistrations and observed no undeclared deviations.

Thank you for taking the time to check both some of the reproducibility of our analyses and the preregistrations. Thank you also for the suggestion, we added it to the ReadMe of the Github repository.

Major comments

R2.3 My main question regards the best way to model the data from experiments 4-7. The manuscript reports two kinds of tests: separate Chi-square tests and GLMs. For the GLMs, there are three models: without expectations, with main effect for expectations, and with interactions with expectation. The manuscript prioritises the results from the Chi-square tests. Depending on the specification of the GLMs, they are inconsistent with the Chi-square tests. For example, for Experiment 5, the Chi-square tests support loss avoidance, but not loss aversion in the PD. This is consistent with the GLM results when controlling for expectations, but in the GLM without expectations. It is not clear to me why Chi-square tests are prioritised in the manuscript; at least for experiments 6 and 7, both are preregistered.

Your next comment (**R2.4**) basically answers this question: because the expectation variable is post-treatment and because it correlates highly with people's actual decisions, interpreting any effect it might have becomes difficult.

The GLM and the Chi-squared results usually agree. The only real differences between GLM and Chi-squared tests appear when adding expectation (as main effect and as interaction), but without expectation, the two are almost always in perfect alignment. Because the interpretation of the GLM tricky for the reason you mentioned, and because the Chi-squared tests don't suffer from this problem, we prioritised the Chi-squared tests.

To clarify this issue, we added the following explanation in the Statistical analysis section (lines 658-664):

For the logistic regressions in Experiments 6 and 7, we preregistered to also include the expectation of what the other person would do as a predictor in the model, both as main effect,

and as interaction effect with loss avoidance and loss aversion. We made the conceptual error that expectation is a post-treatment variable, which makes its addition to the logistic regressions difficult to interpret. As preregistered analyses, we still include the full results, but we prioritise the Chi-squared tests, as these 1) don't suffer from the same problem, and 2) align with the logistic regressions (setting aside the expectation effects).

R2.4 More importantly, the manuscript reports GLMs controlling for expectations. However, unless I misread the manuscript, expectations are post-treatment variables. Controlling for these is problematic because it violates random assignment. I would therefore suggest focusing on GLMs without controls for expectations (if one wanted to control for expectations, some form of strategy method approach would have been preferable).

You are exactly right. Expectation was measured right after participants made their own decisions. It also turns out that expectations are highly correlated with people's own decisions. We only realised our error once all data had been collected. To repeat what we said in response to your comment above (**R2.3**), we therefore de-emphasise the GLM analyses that include expectation as a control variable, because it's not entirely clear what that would mean. However, we did preregister it, so we felt obliged to report the analyses in the manuscript.

Minor comments

R2.5 p. 4: I found the simulations difficult to interpret without knowing the aversion/avoidance parameters. This is explained in the methods, but perhaps it would be possible to add a sentence on p. 4 giving at least some intuition?

Questions about our simulations occurred in all reviewer comments, so we made a few additions (see the Materials & Methods / Simulations section). In response to this comment, we added the following information to page 4 where we introduce the simulations. We hope that this helps readers gain a better intuition about the simulations. The final paragraph (lines 123-129) begins with:

We use two different models to compare loss avoidance (same slope with an intercept) and loss aversion (steeper slope in the loss domain, with no intercept). For loss avoidance, we chose the parameter values $a = 0$, $b = 4$, and for loss aversion we chose $a = 2$, $b = 0$; for the categorical predictions of the simulations, the precise values for each parameter are not critical, within sensible ranges (i.e., in the Prisoner's Dilemma, if loss aversion were extremely large, any category with losses would have a cooperation rate of 0%); instead, the magnitude of the parameter influences the magnitude of the effect (for both loss aversion and loss avoidance).
[...]

R2.6 p. 6: One terminological quibble: you sometimes write "loss avoidance significantly reduced cooperation" (p. 6), or similar phrases. I don't think that's quite accurate -- loss avoidance is what's inferred from the lower cooperation rate. Perhaps a formulation such as "evidence for loss avoidance, indicated by lower cooperation..." would be more accurate.

Thanks for pointing this out, we've made the required changes in phrasing.

R2.7 p. 9: "Stag-Hunt" should be written without a hyphen

Thank you, we corrected it everywhere in the manuscript (we'd always written it with the hyphen, but you're correct, it should be written without).

R2.8 p. 9: "always reduce cooperation" -> "always reduce cooperation in the PD"

Done.

I always sign my reviews.

Simon Columbus | simon@simoncolumbus.com

Thanks again for taking the time to reviewing our article. Your comments have improved the manuscript and we hope we've addressed your comments adequately.

Reviewer #3 (Remarks to the Author):

R3.1 I like the design and evidence in this paper quite a lot. However, the analysis and hence all the tentative conclusions are either clearly not quite correct or not SOTA.

The basic research question is whether affine transformations of payoffs in a 2-person 2-strategy game theory matrix (i.e. multiplying by $a > 0$ and adding b) make a difference to behavior. Specifically, the authors note (building on refs 21-24) that if certain transformed payoffs cross the boundary from positive to negative that could induce changes because of avoidance of losses (a “utility fine”—a great phrase-- $u(x) = x - b$ for $x < 0$) or loss-aversion. This an idea that has been shown in a few experimental economics papers (those carefully cited 21-24) but not picked up in psychology at all. So this is a welcome importation of that idea to a psychology audience, with a much more adventurous multi-game design, and by two talented social neuroscientists who may be able to even more with this idea including in circuitry identification.

Thank you very much for your detailed and valuable comments. We’re assuming that SOTA = State of the art. We hope our responses adequately address your comments.

R3.2 **However, the analysis of the results for some experiments is made complicated by the fact that in experiments 1-3 subjects play repeated PDs multiple times with random (with replacement?) partners in 6-person groups, for 12 payoff matrices. Obviously there are many dynamic influences so these choices, for one person, cannot be treated as iid observations. Test statistics always assume some degree of independence. You need to carefully figure out what statistics to use. For example, players may learn or change over time. The players are also interdependent (e.g. one six person group in which several people always choose D will lead many of them to choose D). So you can’t treat each person in a six-person group as having six different independent cooperation rates. I do not know how to best control for these sources. It is likely that the inflated results from the repeated PD experiments (which do not replicate well in one-shot experiments) are associated with these statistical issues.

a. Frechette (2011) is an old paper on “session effects” (which applies to your 6-person groups as a “session”) and may be helpful as a start.

https://papers.ssrn.com/sol3/papers.cfm?abstract_id=1937814

We agree that this is a tricky issue. Because the iterated Experiments 1-3 include repeated interactions between participants, their responses are not independent from each other. However, after rechecking our analysis and going over alternative ways for correcting for this issue, we believe that the logistic mixed-effects analysis we use for the combined data from Experiments 1-3 already address your concern.

To our knowledge, there are two main ways in which such dependence can be accounted for: the first is to use clustered standard errors (see Cameron & Miller, 2015) in a logistic model. This allows to account for correlations within clusters; in our case, this would be the group. We ran such an analysis in Stata (as in the manuscript on the pooled data from Experiments 1-3):

Logistic regression

Number of obs = **33,629**

Wald chi2(2) = .

Prob > chi2 = .

Pseudo R2 = **0.4043**

Log pseudolikelihood = **-13870.507**

(Std. err. adjusted for **20** clusters in **v2**)

v25	Robust		z	P> z	[95% conf. interval]	
	Coefficient	std. err.				
v18	.0118891	.0172128	0.69	0.490	-.0218474	.0456255
v19	-.1123542	.0333181	-3.37	0.001	-.1776564	-.0470519
v31	2.487775	.2310104	10.77	0.000	2.035002	2.940547

Figure 5 - A cluster-analysis for the pooled data for the iterated Experiments 1-3. V25 = cooperate/defect; v18 = loss aversion; v19 = loss avoidance, and v31 = choice other on previous trial.

If we compare this table to the results from the logistic mixed-effects model (see Table S2), the effects for loss aversion and loss avoidance are very similar, although both effects are slightly smaller in this new analysis (loss aversion was not significant in the mixed-effects model either).

The second approach would be using mixed-effects models. This is what we've included in the manuscript, and we believe that this analysis is more appropriate analysis for our situation: by using nested random effects of Participant in Group in Experiment, we not only correct for the structure of the design (i.e., the correlated behaviour between participants), we actively model the nested structure (see Gelman & Hill, 2007). Thus, we believe that our current analysis already addresses the issue you raise and goes beyond alternative approaches.

To clarify exactly why we ran the logistic mixed-effects model and that it accounts for the issue you raise, we added the following clarification to the section in which we explain the logic for combining Experiments 1-3 into one analysis (lines 200-218):

The results from Experiments 1-3 are thus mixed: Experiments 1 and 3 find evidence for loss avoidance but Experiment 2 does not; Experiment 1 finds evidence for the opposite effect of loss aversion, but neither Experiment 2 nor 3 finds any significant effect of loss aversion. The methodological set-up of Experiments 1-3 was practically identical, so we combined the data to run more comprehensive logistic mixed effects models on the combined dataset from all 114 participants. This analysis was exploratory and not preregistered. We aimed to 1) clarify the role of the null results for loss avoidance from Experiment 2, 2) run hierarchical models by using nested random effects of participant in group in experiment; this corrects for correlated standard errors and explicitly models the nested structure of the experimental design (which the previous analyses do not), 3) test loss aversion in a more nuanced way: our simulations predicted a linear decline in cooperation from Category 1 to Category 5 for loss aversion, so we added Category as an ordered predictor, rather than contrasting Category 1 with Category 5, and 4) ensure that any results for loss avoidance or loss aversion are not confounded by the interactive history between each pair of players: Based on the extensive literature on reciprocity²⁸, we expected that the interaction history would have a large effect on people's decisions. Indeed, across Experiments 1-3, 79% of all decisions were the same as the other person's previous move (generally aligned with the strategy Tit-for-Tat). We therefore also included the other person's previous decision as a predictor to ensure that any effects of loss avoidance or loss aversion were not due to the interaction history.

The paper by Frechette you suggest seems to be in general agreement with what we did. To illustrate this with two quotes:

‘Dynamic session-effects would seem like a potentially more common occurrence, but it is difficult to find many situations where they seem enormous, such as in median type games. This is not to say that session-effects should not be accounted for, however in many experiments it would seem more important to account for the fact that one has repeated observations for each subject.’ (p. 496-7).

And:

‘When one is concerned with the presence of dynamic session-effects, then the appropriate estimator will necessarily depend on the specific source of the problem. As an example, if the issue is that subjects’ behavior in a group is influenced by the feedback they receive about what others did in the previous period, then simply controlling for that could suffice. However, one may suspect the presence of dynamic session-effects without knowing the exact source or the specific form they take, in such cases simply clustering the standard errors may be a reasonable approach, it is not efficient, but more robust’ (p. 496).

We show that we get similar results for both approaches.

The good thing is that, unlike many examples mentioned by Frechette, the source of the potentially problematic session-effect is clear in our case: participants’ choices affect their partners’ choices in subsequent rounds. This we can account for because we know who interacted with whom and how that interaction played out over time. We do this in two ways: first, by using nested fixed effects, and second by using the other player’s previous decision as a fixed effect. In other words, when we run the logistic mixed-effects model, we find that people defect more when doing so can avoid a loss, even when taking into account 1) who is making the decision (player), and accounting for the fact that a participant takes part in a group, which occurs in the context of one of three experiments, and 2) when taking into account whether the other player cooperated or defected in the previous round. Or: given that the other player cooperated or defected in the previous round, and given which participant is taking part in which group and in which experiment, we still find evidence for loss avoidance.

R3.3 A major concern is simple to resolve: How do you get cooperation rates near 50% in the PD for all-positive or all-negative payoffs? The expected value of D is always higher than C for all probabilities p (expected cooperation); so how do you get high predicted cooperation rates? Are you averaging cooperation of one player (e.g. row) computed from expected utilities (lines 117-118) with the assumed 50% of column play? That does not make sense because presumably the column player calculation will be the same as row.

The 50% cooperation rate in the simulations for Categories 1 and 5 were chosen based on a rough estimate of previous empirical literature. Having said that, the precise point of expected cooperation rate in Category 1 is not particularly important because our simulations hold over a wide range of expectation for what the other player will do: as we say in response to comment **R1.3(a)**, our models require some variance, but beyond that the precise numbers are not so important to, it’s more about the patterns we expect to see for each model. We used a sigmoid function to turn value differences between defection and cooperation into probabilities of cooperating. We chose the inflection point of the sigmoid function to be the difference between the defect and the cooperate option, which then leads to a 50% cooperation rate for Category 1.

We could also have chosen an average cooperation rate of 30 or 70%, it wouldn't make much of a difference to the patterns, as long as we don't get to the edges of probabilities where models of our type (that require variance in behaviour) break down. Having said that, the 50% do match our empirical results quite well. To clarify this point, we added the following paragraph to the end of the simulation section (lines 612-622):

Further, we assumed that participants would have a roughly 50% cooperation rate in the 'neutral' (C1, all outcomes are positive) condition in the Prisoner's Dilemma, based on similar previous experiments (and in our experiments, C1 had a roughly 50% cooperation rate in the Prisoner's Dilemma). Formally, to define a 50% cooperation rate for the Prisoner's Dilemma in our simulations, we set the inflection point of the sigmoid function (with beta = 1) to be the difference in expected utility between the D and the C option. As the difference in expected utility is neutral for Stag Hunt and Chicken, this led to a predicted higher overall cooperation rate, which was also borne out by the empirical data. Note that the precise cooperation rate in the neutral condition is not essential for our categorical predictions, as long as it allows for variation (e.g., if the neutral categories had a cooperation rate of % in the Prisoner's Dilemma, loss avoidance could not reduce that any further).

R3.3(a). More generally, it doesn't seem like you have done the proper game-theoretic analysis. In the SOTA analyses of these games, you treat row and column players (since both are human) as making similar calculations (though of course the payoffs may be different) and assuming some p% chance of the other player's behavior. But in the usual benchmark analysis, the value of p is derived endogeneously—i.e. the row player computes utility based on p(column) and column chooses based on p(row). You can then either let p(row) as guessed by the column player be 50% (or some fixed number) but it obviously ignores the fact that the subjects in the experiment may be computing a guess about p from what they think row players are computing.

If we understand your comment correctly, you are concerned that we assume rather than derive the probability of what the other player will do, and that this also neglects potential recursive and strategic reasoning; this departure from game-theoretic analyses might oversimplify how people actually approach such a situation, which limits the results' generalisability.

This comment is similar to **R1.3(a)** in that it questions some of our modelling assumptions. Our goal with the simulations was derive clear and in testable predictions for how loss avoidance and loss aversion would affect people's decisions. To simplify this situation, we assumed a 50:50 expectation of the other person. As we show in response to **R1.3(a)**, deviating from this expectation can change the predictions in extreme cases, but barely affects the predictions when we consider participants' actual (empirical) expectations (see lines 602-611 and Figure S3).

Thus, to make the situation tractable, we simplified the situation, but it seems as if our predictions and conclusions are robust to alternative specifications for the expectation of what the other person will do.

R3.4 The SOTA way to analyze these data is like so, for the one-shot players with no feedback: Start with these lines in the paper

Lines 117-118 contain a nice simple formula for understanding avoidance and aversion (using $U(x)$ to reflect utilities instead of y)

$$U(x|x>0)=x$$

$$U(x|x\leq 0)=ax-b$$

Now loss-avoidance alone is $b>0$.

Loss-aversion is $b=0, a>1$

You can test a hybrid where $b > 0$ and $a > 1$

In one version you assume b and a are the same for everyone and just pile in all the one-shot game data together. Utilities are created above and weighted by a perceived $P(\text{others})$ about what others do. You can set that equal to what others actually do (imposing an equilibrium assumption) or use expectation, or some other idea) (Note that because they're playing once with no repetition the statistical concerns at ** above do not apply at all.) You can then estimate the values of a and b across the population. That would be very interesting. This is called MLE—compute a “likelihood function” $P(\text{cooperate})$ from all those moving parts, and find parameter values (a, b) that maximize how well the prediction fits the data.

We're not entirely sure we understand this comment correctly in its entirety, but if we understand you correctly, you suggest that we should estimate the parameters for loss aversion and loss avoidance over the population. We agree that it would be very interesting to do this, but, in our opinion, this approach is not appropriate for the data in this manuscript. In the one-shot experiments, we only have 5 different payoff matrices. If we were to estimate parameters, this would lead to very imprecisely estimated parameters.

To clarify, we think this is very interesting and the first author is currently working on doing just that for a new project (on the participant-level). But it's not just a matter of estimating parameters to any set of decisions. Estimating parameters like loss aversion and loss avoidance requires many different choices; it requires a different type of data set from the one we collected for this manuscript, data that contains in much finer detail and in much more variation different combinations of loss and gains. With this data, the parameters we get would be largely meaningless, as there would be so much uncertainty about the parameters' true values, that it wouldn't really add any information.

To illustrate that parameter estimates will be imprecise for the data we have, we ran parameter recovery analyses for estimating different levels of slope and intercept (including the linear model of slope = 1 and intercept = 0) for the payoff matrices for which we have empirical data. This is for the example is for the Stag Hunt payoff matrices from Experiment 7, where we have 5 categories of payoff matrices, which means all participants only make five different types of decisions. This leads to a very coarse estimation of utility, which means that many different combinations of slope and intercept fit the data equally well. As the figure below shows, this leads to very noisy parameter recovery:

Figure 6 - A rough illustration of why parameter estimation is not suitable for the dataset we have. Slope = loss aversion, intercept = loss avoidance. The x-axes show the true parameters put into the model, while the y-axes show the estimated parameters using Maximum Likelihood Estimation.

There are many more nuances to be made here, and we're sure the parameter estimation that we offer could be improved even for the data we have by making a few tweaks – but it seems unlikely that this would ever lead to satisfactory accuracy in the parameter estimation. In other words, with the data we collected for this manuscript, we cannot provide any parameter estimates with useful precision.

Our experiments were designed to test categorical difference between predictions, not to estimate parameters. In current work, we're working on this, but for the current data set it would not be appropriate to do parameter estimation.

We hadn't mentioned parameter estimation in our manuscript, but because it is an interesting question, we added it to the discussion (lines 446-452):

A further question for future research lies in estimating the parameters of our model. The current experimental design was optimised for testing whether there was evidence for (or against) loss avoidance. This represents a first step, and future studies could build on our design and optimise the stimuli to be able to estimate the parameters for intercepts, slope, and exponents at the individual level. This would provide a more nuanced perspective into individual variation of loss avoidance (intercept), and its relationship to loss aversion (slope) and diminishing sensitivity (exponent).

R3.5 You also seem to measure “expectations” (e.g. Table S2 refers to it). But methods and Supplemental say nothing about it.

Yes, thank you, we added the following explanation to lines 498-501:

For Experiments 5-7, we also asked participants what they expected the other person to do, ranging from 0 (definitely defect) to 100 (definitely cooperate), but substituting the words

‘cooperate’ and ‘defect’ with the generic labels given to the response options. This question was presented after the experimental decisions.

R3.6 I can't find the Methods/Simulations that are referred to in the reviewer zip file. How exactly were the simulations done?

By this we merely meant the simulation section in the Methods section after the main manuscript. We now realise that 1) we actually call that section ‘Materials & Methods’ and that the specific point at which we link to the further details is somewhat confusing, so we now shifted it to the end of the simulation section in the main text (lines 141-142).

More generally, the details for the simulations can be found in the main text, with additional information (mainly about the assumptions) in the Materials & Methods section. We've updated this section considerably in response to the reviewer comments. The actual code used for the simulations can be found on the linked Github repository (https://github.com/dnhi-lab/losses_gains_2x2/tree/main/simulations).

R3.7 What the supplemental Tables report is mysterious. Take Table S2. What are M1 and M2? What are the percentages in the Table? What test produces the t-statistics? Please be more clear on details

Thanks you for this comment. The table was supposed to provide a quick overview over all tests made, but it seems like it's more confusing than helpful. Since it's not required (Figure 3 in the main text already summarises all results visually), we've decided to just delete both tables (they provide no information that isn't already present in the manuscript elsewhere), and the reference in the discussion to both tables.

R3.8 Materials and methods does not mention anything about stake size

Yes, thank you. Part of this was obscured in Table S1 (*a; see our response to **R3.10**), but it should also be explained in the main text. We added the following lines to clarify the stake size manipulation in Experiment 3 (lines 565-575):

We systematically varied whether the outcomes of the Prisoner's Dilemma games were positive or negative. We took a positive payoff matrix (e.g., in Experiment 1: [T = 7, R = 5, P = 3, S = 2]) and repeatedly subtracted a constant from all payoffs. We then categorized all payoff matrices into one of five payoff matrix categories: Category 1 (C1), in which all outcomes were positive; C2, in which all outcomes were positive apart from S, which was negative; C3, in which T and R were positive, and P and S were negative; C4, in which all outcomes were negative, apart from T, which was positive, and C5, in which all outcomes were negative. In Experiment 3 we also manipulated stake size, using four different levels, with a range of *8 of the smallest stake size. This range was limited by our incentivisation, which added/subtracted up to 5 Euros from the base payment. For all payoff matrices used in all experiments, see Table S1.

R3.9 Line 478. Can you do something with gender and age data? E.g. use them as covariates for predicting behavior. If they do have effects on choice then including them will increase the power to estimate the true effects of loss on behavior.

Our overall approach in this manuscript is to use as few assumptions and degrees of freedom as possible when analysing the data. We want to see whether loss aversion and loss avoidance have

an overall effect on cooperation/decisions in social dilemmas. While we did check for some potential confounds (like the other person's previous decision in the iterated experiments), this was merely done to ensure that the results are due to our intended experimental manipulation and not due to some unexpected confound.

We could in principle add gender and age as predictors, but we're not sure this would be particularly useful. We have no a priori predictions about both of these variables (although there is some evidence in the literature that both could have an effect), which is why we did not preregister any such analyses. Further, we have only limited statistical power to get any reliable results. Thus, for a proper investigation into whether those factors affect loss avoidance or loss aversion, we lack a proper sample, and it would be better to run separate studies designed to address those questions.

R3.10 Table S1 what are the a and b values? Table note does not explain. Is it related to the a and b the in line 118 model?

Thank you for pointing this out. This also relates to your comment about stake sizes (**R3.8**), which was implicit in this table. We've now changed the figure description to clarify what +b and *a mean in the context of creating the different payoff matrices:

Table S1.

All payoff matrices used in the 7 experiments. The payoff matrices were created by taking what we call a base-payoff-matrix, which was always C3, and then adding steps changing +b to reach C1, or subtracting values to reach C5 create the five categories. This way, there is a symmetry across the 5 categories, such that e.g., for the Prisoner's Dilemma the average payoff of Categories 2 and 4 is the same as the average payoff of the Categories 1, 3, and 5, such that when comparing for loss avoidance in the Prisoner's Dilemma, there is no difference in average payoff across that contrast. In Experiment 1, a constant (+b) was added/subtracted from each outcomes such that each of the possible 4 outcomes (T, R, P, S) was positive, 0, and negative at some point. This was changed for the later experiments: we no longer included payoffs with 0, because that was difficult to interpret with respect to loss aversion and loss avoidance (because some payoff matrices include some positive, a neutral, and some negative outcomes), whereas having outcomes as either positive or negative always leads to payoff matrices that can be categorized unambiguously. Multiplying a payoff matrix with *a leads to different levels of stake size (e.g., in Experiment 3, we have four levels of stake size, ranging from 0.5 to 4), without changing the category of the payoff matrix. C1-C5 = Category 1 to Category 5, PD = Prisoner's Dilemma, SH = Stag Hunt, CH = Chicken.

R3.11 The title is not quite right because not all these games involve cooperation (e.g. chicken is mixed-motive).

Thank you, we've changed it to 'Loss avoidance during social interactions'.

R3.12 Lines 159+ these $d=.4$ effects are very large. I suspect you are not estimating them correctly (see comment * above about standard errors)

This is a nuanced point, but we agree with your assessment overall. We rechecked our code and we calculated Cohen's d on the t-statistic (following Rosenthal, 1991), rather than on the means and SDs. While our calculations are correct according to Rosenthal's formula, this calculation on the t-statistic has been shown to inflate Cohen's d (Dunlap et al., 1996). We now changed to the

more conservative original calculation based on the means and standard deviations directly (Cohen, 1988) and reach substantially smaller values for Cohen's d (e.g., the loss avoidance effect for Experiment reduces from $d = 0.433$ to $d = 0.126$, and the loss aversion effect in Experiment 1 reduces from 0.532 to 0.284).

We corrected this in the manuscript and in the online code (calculated in the function `t_test_cooperation`).

R3.13 Lots of nice features of this paper to praise!

a. Line 456 careful Prereg analysis is terrific. That is so helpful for reviewers and it is exactly how Prereg should work, if authors make the extra effort to clarify any deviations. Thank you for that. Figs 1-2 are really good. A lot of information is conveyed there.

Thanks, we're glad you liked those aspects of our manuscript!

References

- Cameron, A. C., & Miller, D. L. (2015). A practitioner's guide to cluster-robust inference. *Journal of human resources*, 50(2), 317-372.
- Cohen, J. (1988). *Statistical Power Analysis for the Behavioral Sciences* (2nd ed.). Hillsdale, NJ: Lawrence Erlbaum Associates.
- Dunlap, W. P., Cortina, J. M., Vaslow, J. B., & Burke, M. J. (1996). Meta-analysis of experiments with matched groups or repeated measures designs. *Psychological methods*, 1(2), 170.
- Gelman, A., & Hill, J. (2007). *Data analysis using regression and multilevel/hierarchical models*. Cambridge University Press.
- Rosenthal, R. (1991). *Meta-analytic procedures for social research* (Rev. ed.). Newbury Park, CA: Sage.

Thanks again for taking the time to review our article. Your comments have improved the manuscript, and we hope we've addressed your comments adequately.

Reviewer comments in blue

Author responses in black

Text from manuscript indented, in different font, with additions and ~~deletions~~.

For easier communication and referencing other comments, we re-numbered all reviewer comments into one coherent scheme (**Rx.y**, where x is the N of the reviewer, and y is the N of comment for that reviewer). Apart from this, all reviewer comments are shown exactly as in the original peer review file.

General remark

We would like to thank all reviewers for taking the time to provide such insightful comments over the course of these rounds of reviews. The final manuscript has certainly improved as a direct consequence of the work you put into reviewing it.

REVIEWER REPORTS:

Reviewer #1 (Remarks to the Author):

R1.1 I think the paper has markedly improved. The exclusion criteria and numbers are transparently presented and my other comments have been addressed.

Thank you very much for reviewing our manuscript (again). We're glad to see that our changes suggested by your comments improved our manuscript.

Reviewer #2 (Remarks to the Author):

R2.1 Thank you for the opportunity to review this revised manuscript. The authors have fully addressed my earlier comments; I have no further objections and believe the manuscript is suitable for publication.

I always sign my reviews.

Simon Columbus | simon@simoncolumbus.com

Thank you very much for reviewing our article (again). We're glad to see that we addressed your comments and that you deem it suitable for publication.

Reviewer #3 (Remarks to the Author):

R3.1 I was asked to comment on the authors' response to Reviewer 3. I will focus here on questions R3.3(a) and R3.4, which seem to me the most fundamental criticisms. In both cases, the authors argue that the proposed approach goes beyond the scope of the paper.

First, comment R3.3(a) suggests an alternative approach to deriving predictions, based not on simulations but on a game-theoretic analysis of belief formation. I agree that this would potentially be interesting, but it strikes me as extending quite a bit beyond the current scope of the paper. Importantly, this would require a commitment to some account of belief formation which wouldn't be testable on the extant data. I think it is fair to stick to the more descriptive simulation approach.

Second, comment R3.4 is, as the authors point out, somewhat cryptic. In my reading, it suggests two things, namely (a) jointly modelling loss aversion and loss avoidance and (b) estimating values for each parameter. I had a similar reaction when I initially read the manuscript -- this is what I would have expected given the question this paper asks. As the authors rightfully argue, this isn't sensible given the data. In this case, I think the manuscript makes a sufficient contribution anyway, even if additional data and modelling could provide further interesting insights. I suspect that Reviewer 3 may not agree with me here, but this seems to me a question more about taste than about accuracy.

I always sign my reviews.

Simon Columbus | simon@simoncolumbus.com

Thank you very much for taking the time to also act as a stand-in for Reviewer 3. Our understanding of your responses is that, while it would be nice to have additional information in our manuscript (like parameter estimates), you agree that this is not feasible with the extant data. Consequently, we did not make any further changes to the manuscript (setting aside the changes we made in the previous rounds of reviews in response to Reviewer 3's comments).